# Higher Resolution, Better Generalization: Unlocking Visual Scaling in Deep Reinforcement Learning

## Abstract

Pixel-based deep reinforcement learning agents are typically trained on heavily downsampled visual observations, a convention inherited from early benchmarks rather than grounded in principled design. In this work, we show that observation resolution is a critical yet overlooked variable for policy learning: higher-resolution inputs can substantially improve both performance and generalization, provided the network architecture can process them effectively. We find that the widely used Impala encoder, which flattens spatial features into a vector, suffers from quadratic parameter growth as resolution increases and fails to leverage the additional visual detail. We test different modifications to the Impala architecture and conclude that, in particular, introducing a global average pooling layer, as in the Impoola architecture, yields consistent improvements across resolutions and network widths while decouples parameter count from resolution—at their respective best conditions, visual scaling unlocks a 28 % performance gain for Impoola over Impala. These gains are strongest in environments that require precise perception of small or distant objects, and gradient saliency analysis suggests that the underlying mechanism is a more spatially localized visual attention of the policy at higher resolutions. Our results challenge the prevailing practice of aggressive input downsampling and position resolution-independent architectures as a simple, effective path toward scalable visual deep RL.

## 1 Introduction

Benchmarks in deep learning go beyond quantifying progress; they actively steer the direction of research itself (Dehghani et al., 2021; Raji et al., 2021; Koch et al., 2021). The structure of benchmarks, e.g., input dimensionality, output format, and data distribution, implicitly determines which architectures are viable and which design choices receive sustained attention. When benchmarks evolve, the methods built around them evolve in response. For instance, the transition in computer vision (CV) from low-resolution datasets like CIFAR-10 (Krizhevsky et al., 2009) to higher-resolution benchmarks like ImageNet (Russakovsky et al., 2015; Deng et al., 2009), was not merely a scaling exercise; it motivated deeper networks (Krizhevsky et al., 2012; Simonyan & Zisserman, 2014; Szegedy et al., 2015), hierarchical feature extractors (Lin et al., 2017), and principled strategies for allocating capacity across layers (Tan & Le, 2019; 2021). As such, resolution is not seen as a superficial preprocessing detail but a first-order constraint that shapes architectural development.

Recognizing this, the CV community has designed architectures that treat resolution as a tunable parameter rather than a fixed constraint. By utilizing global average pooling (GAP) (Lin et al., 2014), models like ResNet (He et al., 2016) and ConvNeXt (Liu et al., 2022) decouple their parameter count from input dimensionality. This flexibility allows models to be pre-trained at standardized scales, e.g., $(224, 224)$, and seamlessly adapted to high-fidelity downstream tasks, e.g., from satellite imagery for land cover classification (Penatti et al., 2015; Helber et al., 2019) to medical diagnostics including chest radiography and dermatology (Rajpurkar et al., 2017; Esteva et al., 2017), without altering the underlying backbone. Research into the train-test resolution discrepancy, such as FixRes (Touvron et al., 2019), further shows that vision models can be fine-tuned across resolutions to improve performance. In CV, resolution has become an adjustable variable while it remains a rigid, largely unexamined constraint in reinforcement learning (RL).

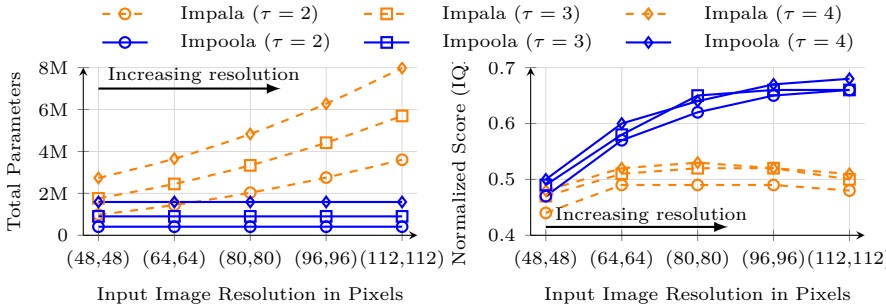

Figure 1: The impact of scaling the input image resolution on the network's total parameter count (**left**) and the performance as normalized score (**right**). We compare the common image encoders Impala (Espeholt et al., 2018) and Impoola (Trumpp et al., 2025) across resolutions from $(48, 48)$ to $(112, 112)$ pixels. Different network widths are shown, i.e., the number of filters per `Conv2d` layer is scaled by $\tau$. The counts include the parameters for separate actor and critic heads. It can be seen that the parameter count in Impala increases with resolution due to its `Flatten` layer, whereas Impoola remains constant due to its `GAP` layer. Given this, we find that Impoola demonstrates superior scaling behavior to Impala, whereas visual scaling unlocks a $28\%$ performance gain for Impoola over Impala at their respective best conditions. These results highlight our proposition that performance gains can be achieved by scaling input resolutions in deep RL.

This rigidity is consequential because visual observations in deep RL are not merely inputs to be classified, but states upon which sequential decisions are conditioned (Mnih et al., 2013; Levine et al., 2016). Therefore, resolution directly affects state aliasing (Whitehead & Ballard, 1991), partial observability (Hausknecht & Stone, 2015), and the precision required for long-horizon control. When agents are trained and evaluated solely on low-resolution abstractions, they risk developing policies that exploit coarse-grained artifacts rather than learning robust spatial features; a form of shortcut learning (Geirhos et al., 2020; Song et al., 2020; Zhang et al., 2018) that undermines transfer to the high-fidelity inputs encountered in real-world deployment (Dulac-Arnold et al., 2019; Ibarz et al., 2021). Yet widely used deep RL benchmarks, including Atari (Bellemare et al., 2013) and Procgen (Cobbe et al., 2020), rely almost exclusively on low-resolution observations. Historically, this was a pragmatic necessity: the computational cost of simulator throughput and the sample-inefficiency of RL algorithms made high-resolution training prohibitively expensive. Environments are routinely downsampled to coarse grids, e.g., $(84, 84)$ for Atari and $(64, 64)$ for Procgen, discarding spatial information and fixing the perceptual scale at which agents operate (Machado et al., 2018).

Because the field has been anchored to low-resolution inputs, standard architectures have been designed around this assumption. The widely used Impala encoder (Espeholt et al., 2018) relies on a flattening operation to transition from convolutional features to the policy head. When resolution increases, this operation causes the first fully connected layer to grow quadratically with spatial input size, concentrating the vast majority of parameters in a single layer; see Figure 1 (left). The resulting severe capacity allocation asymmetry hinders the model from effectively leveraging higher-dimensional inputs. Thus, the low-resolution paradigm has shaped not only evaluation practices but the architectural foundations of deep RL itself. As computational resources and simulator efficiency improve, there is an opportunity to revisit these constraints—yet no systematic study has examined how observation resolution interacts with architecture and learning dynamics in deep RL. Among existing benchmarks, Procgen is particularly well-suited for this study: it is open-source, procedurally generated, computationally lightweight, and offers 16 environments with diverse perceptual demands and built-in easy/hard difficulty modes.

Building this study on a configurable-resolution extension of Procgen, our main contributions are as follows:

- We find in our systematic study strong support that observation resolution should be treated as a first-order variable for policy learning. While the popular Impala encoder mostly fails to leverage additional visual detail, we reveal that minor network adjustments may be sufficient to unlock significant performance gains. In particular, the resolution-independent Impoola architecture, as shown in Figure 1 (right), yields substantial improvement in aggregate performance and general-

ization for visual scaling without any algorithmic changes—we demonstrate this across 16 Procgen environments, five resolutions, three network widths, and multiple training regimes.

- Our results for the hard training regime show that when sufficient performance headroom remains, higher resolution reliably improves both architectures, including standard Impala. Environment-level analysis reveals that the largest gains consistently occur in environments that require precise perception of small or distant entities, where standard resolutions create a perceptual bottleneck.

- We provide mechanistic evidence for these gains through gradient saliency and dormant neuron analysis. Higher resolution enables more spatially focused policy attention on task-critical entities, while Impala tends to disperse its gradient signal and accumulates inactive neurons, presumably because its `Flatten` layer exhibits quadratic parameter growth; see Figure 1 (left).

- Lastly, we introduce Procgen-HD, an open-source extension of the Procgen benchmark that supports arbitrary rendering resolutions while preserving identical game logic, level generation, and reward structure. Procgen-HD enables controlled experiments that isolate the effect of visual fidelity on policy learning, and we release it to facilitate future research on resolution scaling in deep RL.

Overall, we hope that our findings help to challenge the prevailing practice of aggressive input downsampling and that subsequent studies further explore resolution as a tunable variable rather than a fixed constraint.

## 2 Related Work

**Benchmarks and Preprocessing Conventions in Visual Deep RL.** The standard (84,84)-pixel grayscale image that dominates visual RL emerged from pragmatic constraints rather than principled design. Mnih et al. (2013; 2015) introduced this preprocessing for DQN, explicitly noting that the cropping was required because their GPU convolution implementation expected square inputs. The original Atari frames of (210,60) with 7-bit colors were downsampled and cropped to (84,84) grayscale, with 4 frames stacked to handle partial observability (e.g direction of ball in the Pong.) and action repeat of 4 to reduce computational overhead. These choices, born from 2013-era hardware limitations, became unexamined standards that persist across modern benchmarks. Machado et al. (2018) revisited the Arcade Learning Environment (Bellemare et al., 2013) and codified best practices, introducing sticky actions to prevent determinism exploitation. However, this influential work inherited rather than questioned the resolution choice, focusing on stochasticity, episode termination, and evaluation protocols while accepting (84,84) as given. Similarly, Braylan et al. (2015) demonstrated that frame skip significantly affects performance with optimal values varying by game, yet no analogous study examined resolution as a variable.

Subsequent benchmarks inherited these conventions with minor modifications. Procgen (Cobbe et al., 2020) uses (64,64) RGB observations without frame stacking, relying on procedural generation to prevent memorization, but does not justify the resolution choice or provide built-in support for varying it. Similarly, pixel-based DMControl (Tunyasuvunakool et al., 2020) experiments, as standardized by algorithm papers like SAC+AE (Yarats et al., 2021) and DrQ (Yarats et al., 2020), settled on (84,84) RGB with 3-frame stacking, again inheriting from Atari conventions rather than empirical validation. The pattern is consistent: benchmark designers select convenient resolutions that then harden into convention, with reproducibility pressure discouraging deviation.

**Scaling in Vision-Based Deep RL.** Compute scaling through distributed training increased throughput without revisiting visual preprocessing. IMPALA (Espeholt et al., 2018) introduced actor-learner architectures while establishing the Impala model as the de facto encoder. SEED RL (Espeholt et al., 2020) and R2D2 (Kapturowski et al., 2019) further improved throughput and sample efficiency. All these systems kept resolution fixed at (64,64), focusing on sample throughput rather than input fidelity. However, architectural scaling in RL differs fundamentally from supervised learning. Several pathologies explain this: dormant neurons accumulate due to target non-stationarity (Sokar et al., 2023), primacy bias causes overfitting to early experiences (Nikishin et al., 2022), and plasticity loss reduces the ability to fit new targets (Lyle et al., 2022;

Abbas et al., 2023). Kumar et al. (2021) identified implicit under-parameterization, where value networks experience rank collapse despite nominal capacity.

Recent work has identified strategies that address these pathologies. Ceron et al. (2024) demonstrated that mixture-of-experts unlock parameter scaling by reducing dormant neurons, with Willi et al. (2024) extending these findings to multi-task settings. Network sparsity offers another path: Graesser et al. (2022) showed 90% sparsity matches or exceeds dense baselines, while Sokar et al. (2022) found sparse networks learn faster by avoiding memorization of early samples. Most relevant to resolution scaling, concurrent work has identified the flatten operation as a critical bottleneck. Sokar et al. (2025) showed that flattening creates a high-dimensional bottleneck that impedes scaling, whereas tokenization preserves spatial structure. Trumpp et al. (2025) introduced the Impoola encoder, replacing flattening with global average pooling (GAP), further reiterated by the parallel work of Sokar & Castro (2025) on GAP, and Kooi et al. (2025) proposed Hadamax encoding with similar benefits. GAP provides natural resolution independence by reducing the spatial dimensions to a single value per channel, enabling networks to handle varying resolutions without parameter explosion. Similarly, a low-dimensional Linear projection layer also plays an important aspect in the design of DrQ-v2 (Yarats et al., 2022).

## 3 Experimental Setup

This work analyzes the relationship between performance and increased input resolutions alongside model capacity. We describe our benchmark below and provide details on the network design and training settings.

**Benchmark.** This work's analysis is based on the Procgen Benchmark (Cobbe et al., 2020), a suite of 16 procedurally generated environments that provides a more robust evaluation of agent capabilities compared to the fixed environments of the Atari suite. While Atari games rely on a static set of textures and deterministic transitions, Procgen environments feature high-variance procedural generation, ensuring that agents encounter unique visual combinations of layouts and assets in every episode. This structural and visual diversity provides a clearer signal for evaluating the impact of input resolution, as the agent must resolve fine-grained details in constantly changing environments rather than relying on fixed pixel patterns.

To systematically investigate the effects of scaling, we use our custom modification of the benchmark, *Procgen-HD*, which supports configurable rendering resolutions. Formally, let the observation $x \in \mathbb{R}^{R \times R \times 3}$ denote a tensor of an RGB image with $(R, R)$ pixels, i.e., the resolution is $\mathrm{d}(x) = (R, R)$. We evaluate performance across the discrete set of resolutions $R \in \{48, 64, 80, 96, 112\}$, covering a spectrum of visual fidelity with respect to the original standard resolution of $R = 64$ (Cobbe et al., 2020). We do not use frame stacking or grayscale conversion, adhering to the standard Procgen protocol. Figure 2 visualizes example images from four Procgen-HD environments, with the remaining 12 games detailed in Appendix A.1. Lower-resolution renderings exhibit severe visual artifacts that obscure key details, e.g., determining the orientation of the green ego fish in *Bigfish* or the ego aircraft in *Starpilot* becomes challenging Crucially, increasing the resolution in Procgen-HD does *not* expand the field of view but solely increases the information density.

**Network Architecture.** For image processing, all experiments use an identical ResNet-based backbone derived from the Impala architecture (Espeholt et al., 2018), which has been widely adopted as the standard in recent state-of-the-art approaches. As visualized in Figure 3, this backbone processes the input observation $x$ through a stack of three hierarchical convolutional sequences `ConvSeq`. Each sequence consists of a `Conv2d` layer with stride=1, followed by a max-pooling operation with stride=2 and a series of two residual blocks `ResBlock`. The `ResBlock` units consist of two `Conv2d` layers with stride=1. For all `Conv2d` layers in each of the three `ConvSeq`$_{\{0,1,2\}}$, the base configuration uses the same number of $\{16, 32, 32\}$ filters, respectively, each with a kernel size of $(3, 3)$. We allow scaling the number of filters per layer width multiplier $\tau$, following prior work on scaling the Impala backbone (Cobbe et al., 2020; Trumpp et al., 2025), allowing for varying the representational capacity while keeping the backbone structure constant. Since the results of these works showed the effectiveness of width scaling, we only examine already scaled networks with $\tau \in \{2, 3, 4\}$.

**Visual Scaling Dynamics.** Let $H \times W$ be the spatial dimensions and $C$ the number of channels of the convolutional feature map $m \in \mathbb{R}^{H \times W \times C}$ output by the last `ConvSeq`$_2$. Given the max-pooling operations (stride=2), the backbone yields a total spatial downsampling factor of $k$, resulting in $H = W = \lfloor R/k \rfloor$, e.g.,

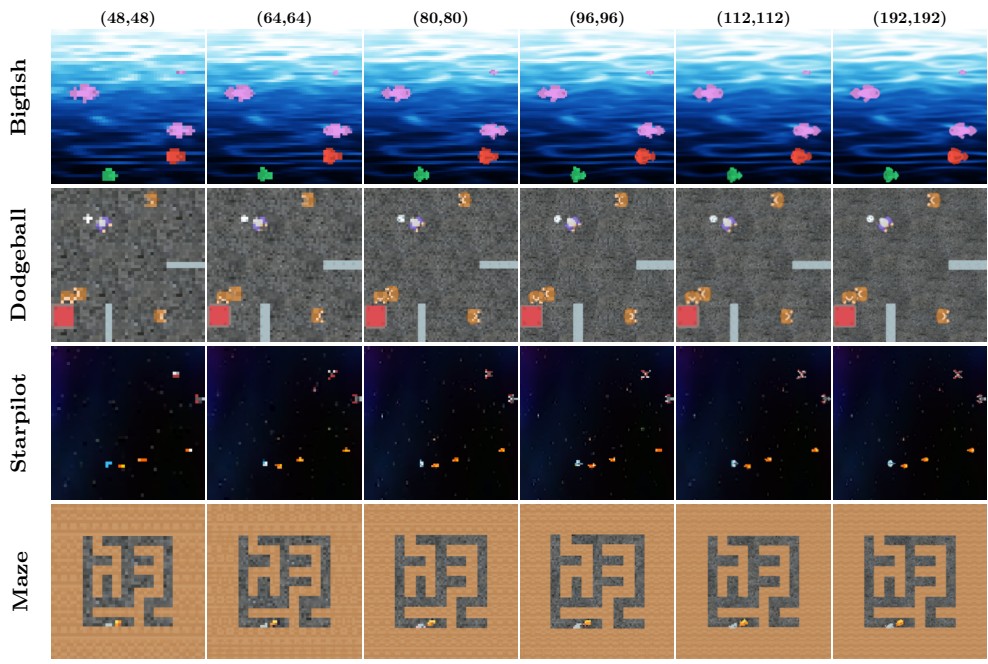

Figure 2: Comparison of a subset of 4 Procgen-HD environments at different image resolutions of $(R, R)$ pixels. All images depict the same scene, rendered at varying resolutions $R \in \{48, 64, 80, 96, 112, 192\}$ with the *field of view remaining constant.* The $(192, 192)$ resolution is provided solely for visual comparison against an image with minimal compression artifacts but not further evaluated.

$k = 8$ in the base configuration. Consequently, when resolution $R$ scales, the feature map dimensions $H, W$ increase, meaning that the relative coverage of the fixed absolute receptive field $r_{\text{field}}$ spans a progressively smaller fraction of the input image[1]. Both the original version of Impala and the recently proposed derivative Impoola share this backbone and map the final feature map $m$ to a latent representation vector[2] $z \in \mathbb{R}^{\text{d}(z)}$ via a `Linear` projection layer $z = \text{Linear}(e) = W_{\text{proj}}e + b$, where $e$ is an intermediate feature vector. However, the variants differ fundamentally in how they transition from spatial feature maps $m$ to this vector $e$:

- **Impala (Flatten)** (Espeholt et al., 2018): The standard approach flattens the spatial feature map $m$ directly by unrolling the spatial and channel dimensions into a single vector $e \in \mathbb{R}^{H \cdot W \cdot C}$. Crucially, even if the projection output dimension $\text{d}(z)$ remains bounded, the projection layer's weight matrix

$$W_{\text{proj}} \in \mathbb{R}^{\text{d}(z) \times ((R/k)^2 \cdot C)} \tag{1}$$

  scales quadratically with the input resolution $R$. This property causes a massive increase in the parameter count of the `Linear` projection layer, directly linking model size to visual fidelity.

- **Impoola (GAP):** As proposed by Trumpp et al. (2025), this variant applies global average pooling (GAP) after the final `ConvSeq`$_2$, similar to Sokar & Castro (2025). GAP aggregates the spatial layout by averaging over $H \times W$, outputting a fixed-size feature vector $e \in \mathbb{R}^C$. The weight matrix

$$W_{\text{proj}} \in \mathbb{R}^{\text{d}(z) \times C}, \tag{2}$$

  of the subsequent `Linear` projection layer yields a static parameter count, thus effectively decoupling the parameter count from the input resolution, ensuring the model size remains *constant* even as visual fidelity increases. In addition, the GAP operation introduces a strong information bottleneck since it removes all spatial coordinate information entirely prior to projection.

---

[1]As derived in Appendix B.4, the baseline configuration maintains a constant absolute receptive field of $r_{\text{field}} = 141$. This translates to a relative coverage of $\approx 220\%$ at a standard resolution of $R = 64$, which shrinks to $\approx 125\%$ at $R = 112$.

[2]The latent vector $z$ serves as input to the downstream network heads, e.g., the separate actor and critic network heads; unless otherwise stated, we use $\text{d}(z) = 256$, which is in line with other works.

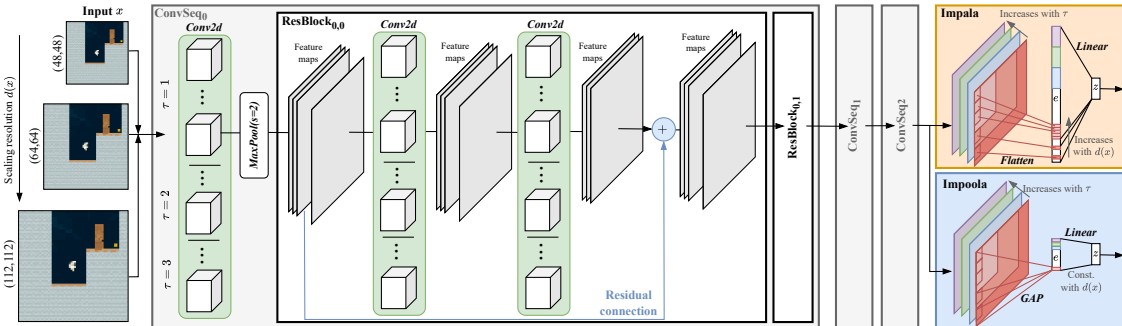

Figure 3: As image resolution $d(x) = (R, R)$ increases, the `Flatten` layer in Impala (**top**) leads to parameter explosion of the `Linear` projection layer $z = W_{proj}e + b$ since the dimension $d(e)$ increases. In contrast, the Impoola architecture (**bottom**) fully mitigates this situation because global average pooling (GAP) maintains a constant representation size $d(e)$. Figure derived with permission from Trumpp et al. (2025).

**Training Settings.** Unless otherwise stated, our results encompass the *full* benchmark across all 16 environments. Following the established standard (Cobbe et al., 2020), we train for 25M timesteps in the *easy* setting and 100M timesteps in the *hard* setting.

We primarily focus on the generalization track, which allows us to measure generalization by defining distinct sets of levels for training and testing. The *easy* setting restricts training to 200 simpler levels, whereas the *hard* setting uses 1000 levels with significantly increased game difficulty and complexity. The additional results for the *efficiency* track do not restrict the number of training levels.

**DRL Algorithm.** As proximal policy optimization (PPO) (Schulman et al., 2017) is the established baseline for Procgen, maintaining this standard allows us to isolate the impact of input resolution and architectural scaling without the confounding influence of algorithmic choice. The actor and critic for PPO share the image encoder; further implementation details and hyperparameters are listed in Appendix A.2.

**Evaluation Metrics.** To measure generalization capability in the generalization track, we evaluate performance on the held-out test levels that the agent has never encountered during training. The efficiency track is evaluated on all levels. During training, we periodically collect episodic returns over 2,500 episodes using these unique levels. These returns are normalized to obtain scores using the constants from Cobbe et al. (2020), where 1.0 denotes optimal performance and 0.0 corresponds to a random policy. Each environment is evaluated with 5 independent runs using different random seeds. Aggregated results across environments are reported as interquartile mean (IQM) (Agarwal et al., 2021) scores, with 95% stratified bootstrap confidence intervals shown as shaded regions.

## 4 Empirical Results

We argue that observation resolution is a first-order variable that has been held fixed by convention rather than by design, and that this has left both architectural limitations and potential gains unexplored. We test this hypothesis empirically, asking whether increasing resolution improves performance and which architectures can leverage it (**RQ1**), whether these gains reflect genuine generalization (**RQ2**), whether the findings hold across training regimes (**RQ3**), and which perceptual demands drive them (**RQ4**).

### 4.1 RQ1: Does Higher Resolution Improve Performance?

As initially shown in Figure 1 and detailed in Figure 4, the effect of increasing resolution depends strongly on architecture. Impala shows a modest improvement from $(64, 64)$ to $(80, 80)$, but this trend does not persist, and performance declines at higher resolutions across width scales, suggesting that additional visual detail is not effectively utilized. Notably, this improvement emerges only at $\tau = 3$, whereas lower capacity Impala variants do not benefit from increased resolution. In contrast, Impoola improves aggregated scores

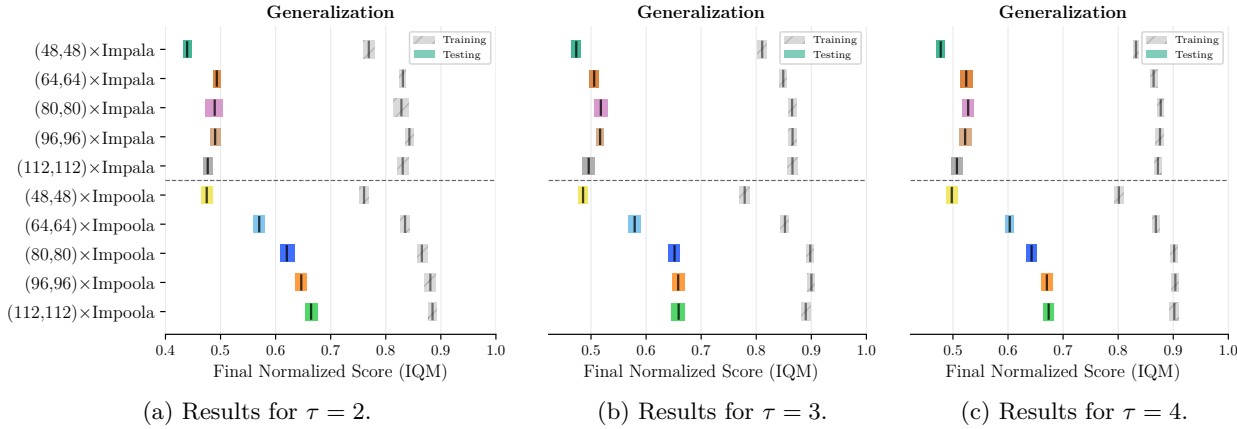

(a) Results for $\tau = 2$.      (b) Results for $\tau = 3$.      (c) Results for $\tau = 4$.

Figure 4: Comparison of the final aggregate results after $25\,\mathrm{M}$ training steps for the *easy* generalization across width scale $\tau \in \{2, 3, 4\}$ and image resolutions of $(48, 48)$ to $(112, 112)$. Evaluation of the training levels (gray) is shown alongside performance on held-out test levels (colored).

consistently with resolution across all capacity configurations. The best aggregate testing score of 0.674 is achieved by Impoola with $\tau = 4$ at $(112, 112)$, representing an approximate $18\,\%$ improvement only through scaling in comparison to itself at the standard configuration with $(64, 64)$ and $\tau = 2$. Impala's best score of 0.527 is achieved with $\tau = 4$ at $(80, 80)$, benefiting mostly from the increased network size with $\tau = 4$ rather than the higher image resolution. Ultimately, under their respective best conditions, Impoola outperforms Impala by $28\,\%$.

We suspect that the performance difference arises from how each architecture handles the growth of spatial feature maps $m$. As resolution increases, Impala produces larger feature maps whose flattened representation expands rapidly, limiting its ability to effectively exploit the additional spatial information. Impoola, by contrast, remains largely size-independent through the `GAP` layer and therefore adapts seamlessly to higher resolutions, as previously summarized by Figure 1. Note that increasing R also influences the coverage of the input image by the network's receptive field. However, we calculate in Appendix B.4 that the theoretical absolute receptive field in Impala is $r_{\mathrm{field}} = 141$, which is high enough to still fully cover the image at R=112.

Consequently, a natural question is whether increasing network depth can improve Impala. Adding a fourth `ConvSeq` block increases the model depth to 20 layers, thereby increasing the downsampling factor to $k = 16$, which limits the growth of the flattened representation $e$, but also increases the absolute receptive field to restore a higher coverage rate of the input image; see Appendix Table 2. As shown in Figure 5, the deeper Impala model clearly benefits from increased resolution. While prior Procgen scaling work (Cobbe et al., 2020) has primarily emphasized width, i.e., increasing the number of filters per `Conv2d` layer, these results suggest that depth provides a complementary axis for enabling effective resolution scaling.

We also analyze further variations of the Impala architecture that aim to address the absolute receptive field coverage or the parameter explosion in the `Linear` layer in Appendix B.1.2. We find that, in particular, reducing the dimensionality of the `Linear` projection layer improves performance, showing that gains can also be achieved with other modifications. With respect to this, we suspect that the benefit of the network with $4\times$`ConvSeq` is rather related to the `Linear` layer and reduced dimensions than the increased receptive field. Moreover, we discuss in Appendix B.1.1 that a recent tokenization-based architecture (Sokar et al., 2025) also exhibits substantial gains from resolution scaling. However, we conclude that Impoola's superior results position the `GAP` layer as the preferred modification, combining the best performance with practical benefits such as a simple implementation and a low parameter count.

## 4.2 RQ2: Does Higher Resolution Improve Generalization?

While RQ1 showed that higher resolution improves aggregate performance, we next ask whether these gains reflect genuine generalization or simply stronger fitting to the training levels.

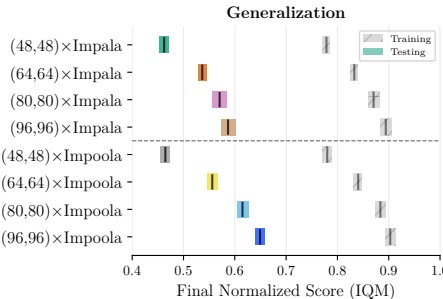 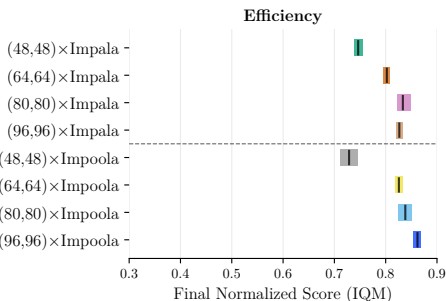

Figure 5: Aggregate results of testing levels in the generalization track for *deeper* networks, which have a fourth `ConvSeq` block with 32 filters in the unscaled setting added, totaling 20 layers. Results are shown for a width scale of $\tau = 3$.

Figure 6: Aggregate testing results for the efficiency track with a width scale of $\tau = 3$. The efficiency track removes the level restriction entirely, exposing agents to the full procedural distribution during training.

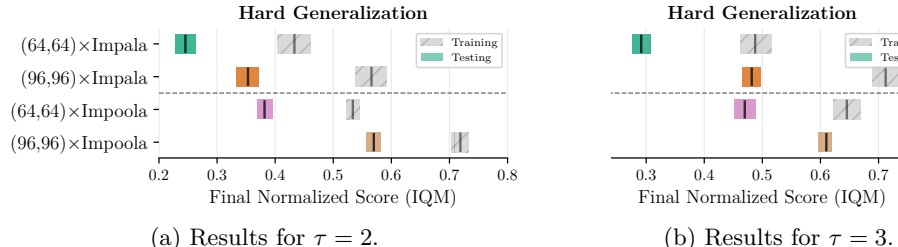

(a) Results for $\tau = 2$.

(b) Results for $\tau = 3$.

Figure 7: Comparison of the final aggregate results after $100\,\mathrm{M}$ training steps for the *hard* generalization across width scale $\tau \in \{2, 3\}$ and input image resolutions of $(64, 64)$ and $(96, 96)$. Evaluation of the training levels (gray) is shown alongside performance on held-out test levels (colored).

In the easy generalization setting, where training is restricted to 200 levels, agents achieve near-optimal training scores of approximately 0.9 on these levels across most configurations. This training saturation compresses the visible benefit of higher resolution in the aggregated results, as improvements can only manifest on unseen test levels. To isolate the effect of resolution on generalization, we examine training and test performance separately.

Figure 4 shows that the relationship between resolution and generalization differs across architectures. For Impala, increasing resolution yields only small improvements in training performance, while test performance declines at $\tau = 2$ and improves only slightly at $\tau = 3$, leaving a persistent gap between training and test results. In contrast, Impoola exhibits a different pattern. Training performance changes only modestly with increasing resolution, whereas test performance improves substantially, narrowing the gap between them. This indicates that even small improvements during training translate into stronger gains on unseen levels. For $\tau = 4$, Impoola's train–test gap at $(48, 48)$ is approximately 0.3, which reduces to 0.23 at $(112, 112)$, consistent with improved generalization at higher resolution. Even within this saturating regime, Impoola's consistent gains confirm that the standard $(64, 64)$ rendering discards visual detail that, when preserved, measurably improves generalization.

### 4.3 RQ3: How Robust Are These Findings Across Training Regimes?

To determine how the impact of resolution depends on the training regime, we evaluate two additional Procgen-HD tracks that vary the task difficulty and data diversity.

**Hard Generalization Track.** We evaluate $\tau = 2$ and $\tau = 3$ at $(64, 64)$ versus $(96, 96)$. As shown in Figure 7, increasing resolution leads to clear performance improvements for both architectures. With 1000 training levels and substantially harder tasks, agents operate further from optimal performance, allowing

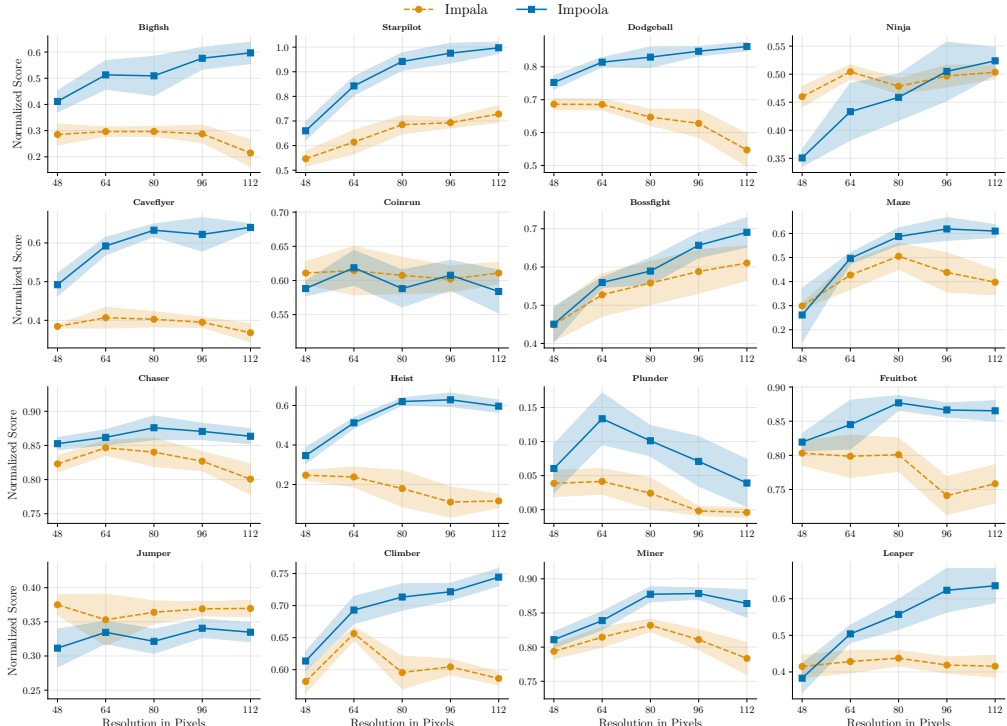

Figure 8: Environment-level comparison for all 16 Procgen-HD games for generalization, showing the final normalized scores of Impala and Impoola for testing levels. Both architectures use a width scale of $\tau = 4$; the resolution increases from $(48, 48)$ to $(112, 112)$.

additional visual detail to translate more directly into improved decision quality. Increasing network width further strengthens these gains. Although Impala continues to exhibit a larger train–test gap than Impoola, it still benefits substantially from higher resolution, while Impoola consistently converts increased visual fidelity into stronger performance across both width scales. Overall, these results indicate that when sufficient headroom remains, higher resolution reliably improves performance across architectures.

**Efficiency Track.** The efficiency track removes the level restriction entirely, exposing agents to the full procedural distribution during training. As shown in Figure 6, the gains from increasing resolution are more modest in this setting, which is expected: unlike the generalization track, i.e., training is restricted to 200 levels, agents now encounter the full level diversity during training, leaving less room for resolution to improve coverage of unseen variations. Nevertheless, Impoola continues to benefit from higher resolution and outperforms Impala on average across resolutions, supporting that the resolution gains observed in the generalization setting reflect genuine perceptual improvements rather than artifacts of limited training diversity. Reducing resolution below $(64, 64)$ degrades performance for both models, confirming that the standard resolution represents a meaningful threshold below which critical visual information is lost.

### 4.4 RQ4: Which Environments Benefit Most from Higher Resolution?

The aggregate results show that resolution scaling improves performance, but aggregate metrics alone do not reveal where these gains originate. To identify which environments benefit most from higher resolution, we analyze environment-level performance in Figure 8

The largest gains occur in environments requiring precise perception of small or distant entities. In *Dodgeball* and *Starpilot*, where the agent must accurately perceive and track relatively small, moving objects, increasing resolution from $64 \times 64$ to $112 \times 112$ yields substantial performance leaps. Similarly, in *Maze* and *Heist*,

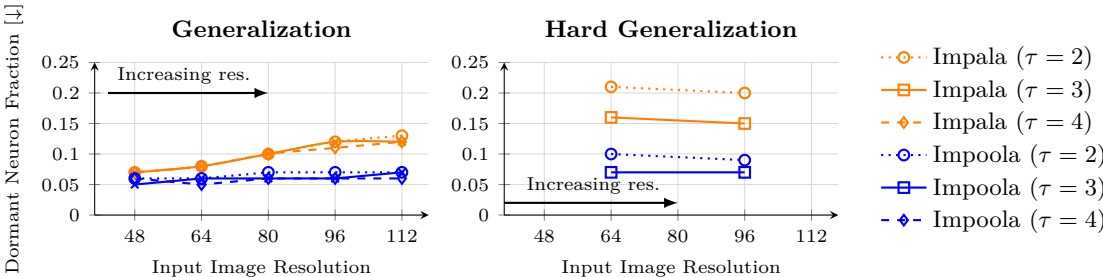

Figure 9: Comparison of dormant neurons for generalization with the easy (**left**) and hard (**right**) setting.

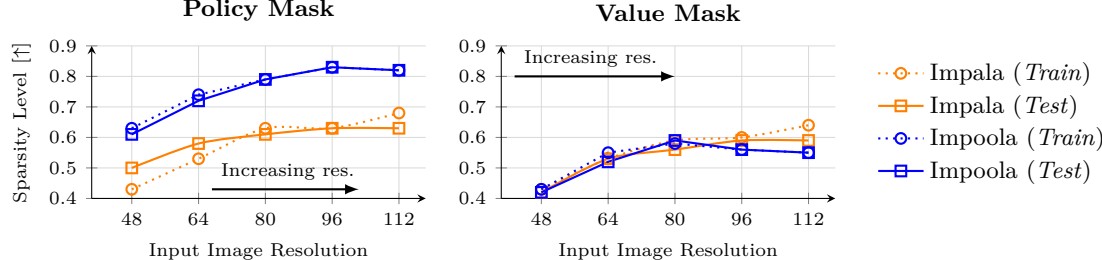

Figure 10: Comparison of policy (**left**) and value (**right**) mask sparsity, based on saliency maps and networks with a width scale of $\tau = 2$ for the generalization track. Saliency maps are calculated for observations obtained for the restricted testing and training levels, calculated as the mean over all observations encountered in 5 full evaluation episodes, and using a threshold of $\epsilon = 0.1$ for masking.

higher resolution likely improves visibility of distant structural cues and key objects, enabling more efficient navigation. In these environments, perceptual fidelity is a clear bottleneck at standard resolution.

In contrast, environments dominated by large static structures or simple locomotion, such as *Coinrun*, *Chaser*, and *Jumper*, show diminishing returns from resolution scaling. Because the standard $(64, 64)$ resolution already resolves critical obstacles, improvements remain modest, indicating that difficulty in these environments is driven more by dynamics than perception. It is noteworthy that *Jumper* and *Coinrun* are games with *agent-centered* observations, so these results overlap with the findings from Trumpp et al. (2025), who discuss that Impala often possesses an initial, advantageous inductive bias for such games in comparison to Impoola. However, as demonstrated by the trend in *Ninja*, our findings show that scaling the input resolution may eventually help Impoola outperform Impala for such games as well.

Additionally, while resolution scaling does not necessarily resolve general convergence issues, e.g., as found for Plunder, the trend towards improved performance from resolution scaling is significantly intensified in the *hard* generalization setting. As seen in Figure 21, the increased difficulty makes higher visual fidelity advantageous across *all* environments for Impoola, and it also benefits Impala in the majority of games.

## 5 Representational Analysis

We examine *why* Impoola benefits more from higher resolution than Impala by analyzing two properties of the learned networks: the fraction of dormant neurons (Sokar et al., 2023), which measures how broadly each architecture utilizes its capacity, and gradient-based saliency (Wang et al., 2016), which reveals where in the visual input the network directs its attention.

### 5.1 Dormant Neurons

In the easy generalization track shown in Figure 9 (left), Impala exhibits a clear monotonic increase in dormant neurons as resolution grows. Despite the network gaining additional capacity at higher resolutions,

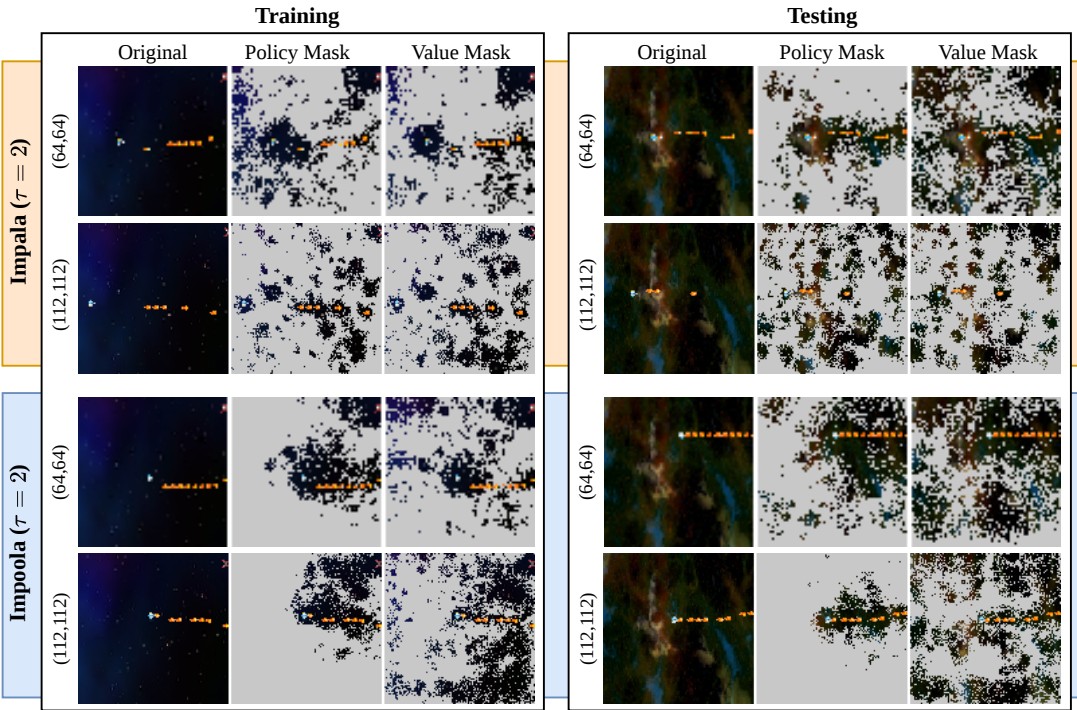

Figure 11: Qualitative comparison of saliency-based masks for the *Starpilot* game. Masks are binary overlays generated where the normalized saliency map exceeds a threshold of $\epsilon = 0.1$. The visualization highlights the specific regions retained by the Policy and Value heads for the Impala and Impoola agents, demonstrating the sparsity of the input processing at varying image resolutions. Pixels that are below the masking threshold are visualized as gray areas.

a progressively larger fraction of neurons remains inactive, indicating that this capacity is not effectively translated into richer representations. In contrast, Figure 9 (right) reveals a different pattern for the hard generalization track: dormant neuron fractions slightly decrease with resolution, suggesting that higher task complexity encourages broader use of the network's capacity. Notably, these trends are consistent with the performance results in each setting, where resolution gains are larger in the hard track.

Across both settings, Impoola consistently maintains lower and more stable dormant neuron fractions, with the gap becoming particularly pronounced in the harder track. This suggests that GAP-based architectures better preserve active representations as resolution increases, rather than accumulating unused capacity. We provide a per-layer analysis of dormant neurons in Appendix B.1.3, where we find that, in particular, the first `ConvSeq`$_0$'s `ResBlock` layers exhibit high relative dormant neuron counts; we suspect that, when the `Linear` layer's parameter count in Impala explodes, gradient issues arise in these early layers. While the relative dormant neuron count in this `Linear` layer is similar to Impoola, it means that the absolute count is still substantially higher due to the disproportionate parameter allocation to it in Impala, see Appendix C

## 5.2 Saliency Maps

The dormant neuron analysis characterizes how each architecture utilizes its internal capacity, but does not reveal *where* in the visual input the network directs its attention. To investigate this, we employ gradient-based saliency analysis, computing the gradient of the value function $\nabla_x V(x)$ and the policy logit $\nabla_x \log \pi(a \mid x)$ with respect to the input observation $s$, following saliency visualization methods used for deep RL agents (Wang et al., 2016). For each frame, we take the absolute gradient magnitude, normalize it to $[0, 1]$ using 99th-percentile clipping, and apply a binary threshold mask at $\epsilon = 0.1$: pixels whose normalized gradient exceeds $\epsilon$ are marked as attended, while the remainder are suppressed. We summarize these masks with a scalar *mask sparsity* metric, defined as the fraction of pixels below the threshold. Higher sparsity

indicates that the network concentrates its gradient signal on a smaller, more localized region. We conduct the following analysis on *Starpilot*, where the agent must track small, fast-moving projectiles and enemies across diverse backgrounds, and where resolution scaling yields a substantial performance gain.

As shown in Figure 10 (right), the value mask sparsity remains relatively flat across resolutions for both architectures, and the corresponding masks in Figure 11 show broad activation covering large portions of the scene. This is expected: $V(x)$ captures a global assessment of the state, so its gradient signal naturally spreads across the full observation rather than localizing on specific entities. The policy mask sparsity presented in Figure 10 (left) reveals a contrasting pattern. Impoola's sparsity increases steadily from approximately 0.72 at $(64, 64)$ to 0.82 at $(112, 112)$, meaning the network attends to a progressively smaller fraction of the input as resolution grows, concentrating tightly on the agent, nearby enemies, and projectiles while suppressing background regions; see Figure 11. Impala's policy sparsity, by contrast, remains largely constant on test levels, and its masks stay diffuse even at higher resolutions.

We reason that at low resolutions, small objects are obscured by blurring and aliasing, whereas higher resolution restores the visual distinctness required for precise localization. However, the fact that this increased sparsity appears in Impoola but not in Impala suggests that the architecture needs be able to process the additional spatial detail effectively for it to translate into more focused policy attention.

## 6 Limitations

A central challenge in studying resolution scaling is the high computational cost of processing high-dimensional inputs. Increasing the observation resolution from $(48, 48)$ to $(112, 112)$ results in a substantial increase in floating-point operations and memory usage. We visualize this cost in Appendix Figure 23, e.g., increasing from Procgen's standard resolution of $(64, 64)$ to $(112, 112)$ results in an approximately $2.7\times$ longer total training time; Impoola and Impala have the same training times since the cost of the Linear layer is negligible compared to the Conv2d layers. As such, this intense computational demand necessitated a focused scope; our study's results are already based on more than 20,000 A100 GPU hours for training. We therefore ground our analysis in the Procgen benchmark, specifically adopting the PPO algorithm and the Impala and Impoola models in accordance with the benchmark's established standards.

Our choice of the Procgen benchmark offers particular advantages over traditional suites like Atari, as it explicitly evaluates generalization across a diverse set of games with varying visual characteristics, enabling robust assessment of an agent's perceptual capabilities. However, since our study is conducted entirely within this procedural, discrete-action domain, it should be seen as a first compelling demonstration of visual scaling laws at higher resolutions, but further exploration is required. As such, we hope the community will extend our effort by investigating continuous control tasks, 3D environments, and off-policy methods in the future.

## 7 Conclusion

This work challenges the prevailing assumption that low-resolution inputs are sufficient for deep RL. Using Procgen-HD, a configurable-resolution variant of the Procgen benchmark, we show the potential of higher resolutions to improve both performance and generalization. Specifically, we find the biggest improvement when leveraging the resolution-independent Impoola architecture that uses `GAP`. However, gains can also be realized with other modifications, e.g., a reduction of the bottleneck dimension or SoftMoE tokenization. We suspect that the standard Impala encoder stagnates at higher resolutions due to quadratic parameter growth in its `Flatten` layer, while Impoola's `GAP` layer decouples capacity from resolution and achieves a 28 % improvement at their respective best conditions. Environment-level analysis reveals that these gains concentrate where precise perception of small or distant entities matters most, and gradient saliency analysis implies that a key mechanism is an improved spatial localization in the policy network. These findings suggest that the conventional $(64, 64)$ and $(84, 84)$ defaults are not architecturally neutral but instead impose a perceptual ceiling that favors resolution-dependent designs and limits scalability. We hope these initial findings encourage broader study of the impact of resolution scaling in visual deep RL. We also see particular relevance for the robotics community, which often deals with high-resolution images.

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

# A  Experiment Details

## A.1  Procgen-HD

Procgen-HD is a custom extension of the Procen Benchmark which allows for native rendering at different resolutions. There are the same 16 games available in Procgen-HD: Figure 2 shows *Bigfish*, *Dodgeball*, *Starpilot*, and *Maze*. The remaining 12 games are visualized in Figures 12 and Figure 13, respectively.

### A.1.1  Observation Renderings

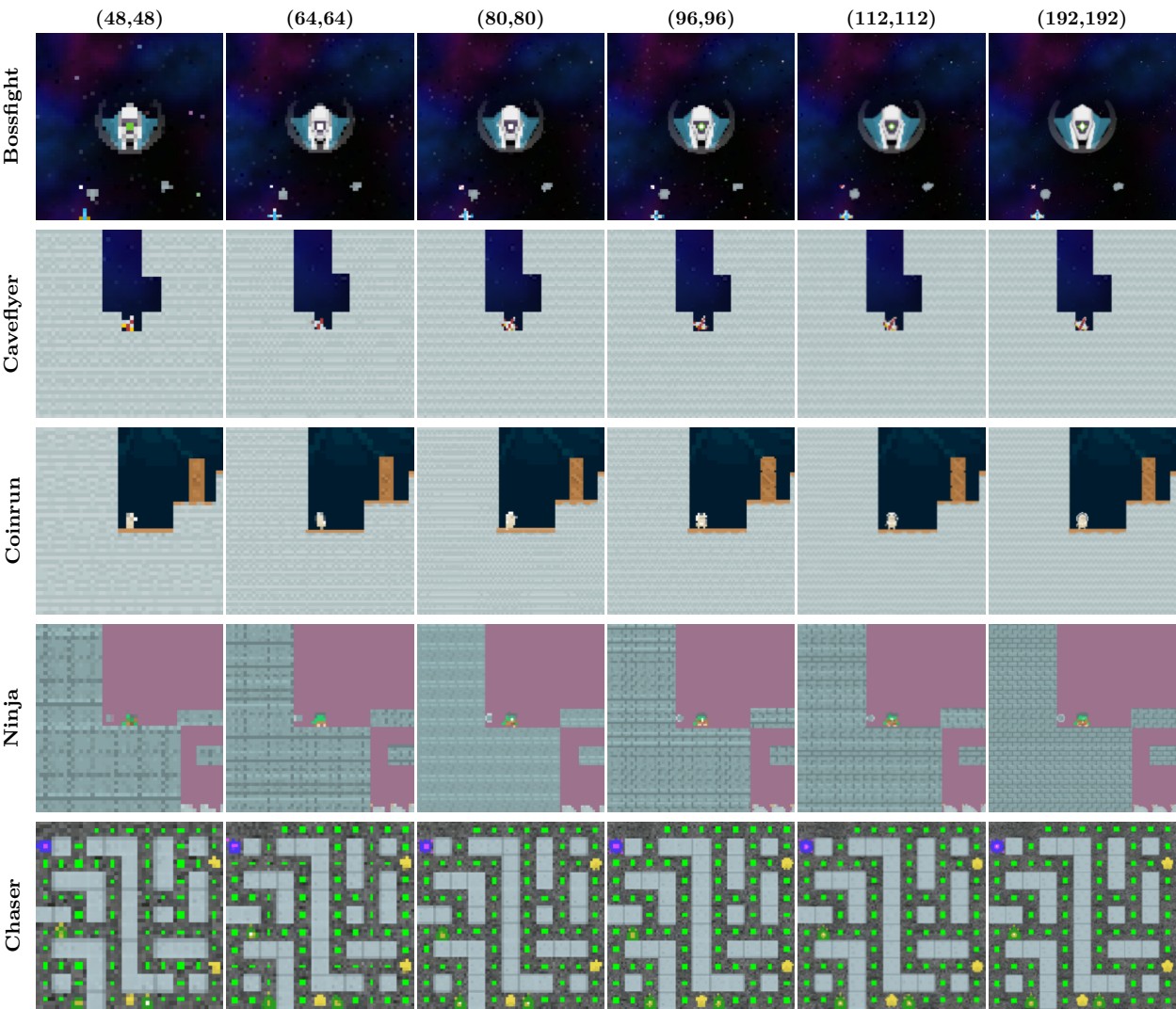

Figure 12: Comparison of further Procgen-HD environments at different image resolutions of $(R, R)$ pixels. All images depict the same scene, rendered at varying resolutions $R \in \{48, 64, 80, 96, 112, 192\}$ with the field of view remaining constant. The $(192, 192)$ resolution is provided solely for visual comparison against an image with minimal compression artifacts but not further evaluated.

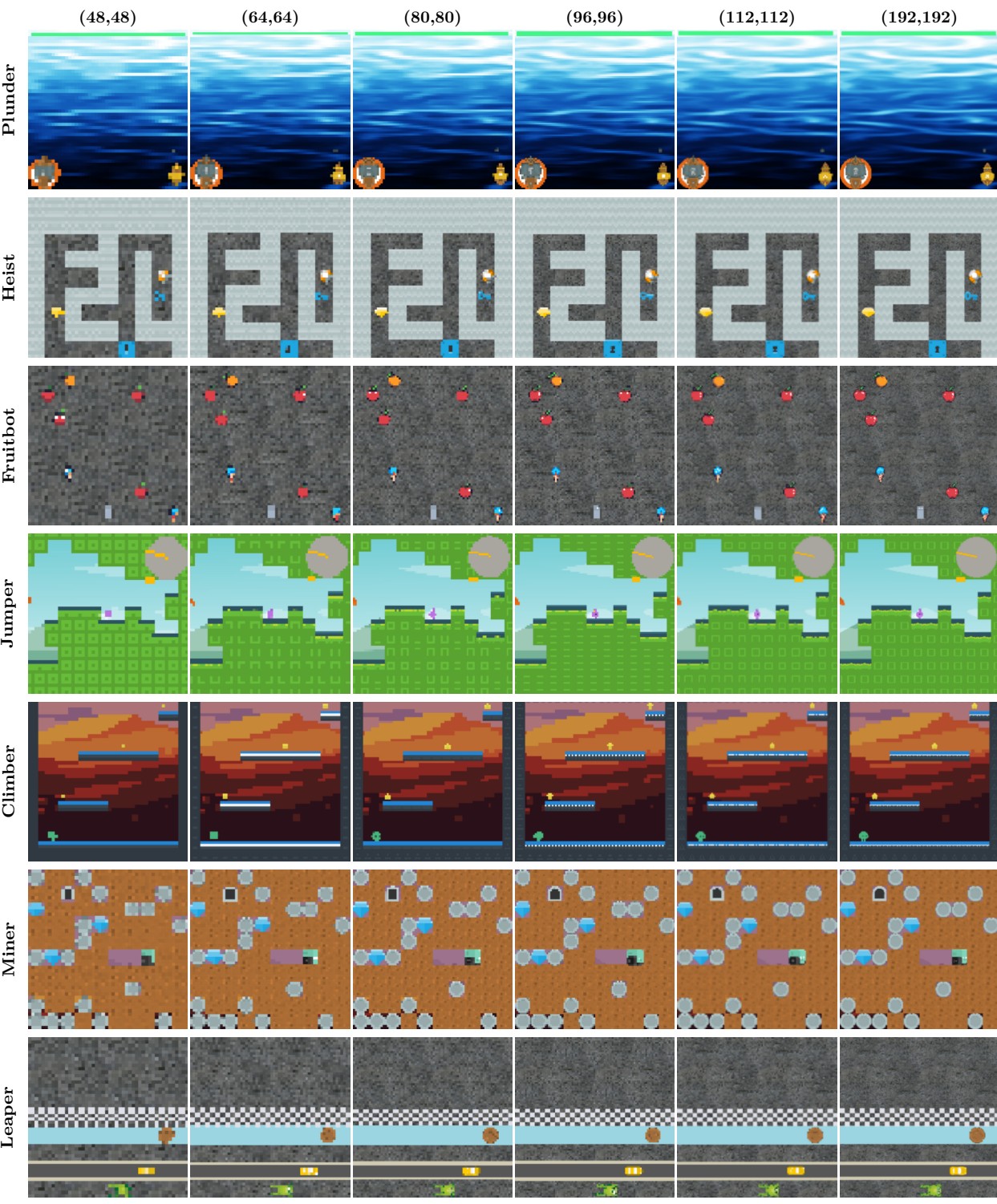

Figure 13: Comparison of further Procgen-HD environments at different image resolutions of $(R, R)$ pixels. All images depict the same scene, rendered at varying resolutions $R \in \{48, 64, 80, 96, 112, 192\}$ with the field of view remaining constant. The $(192, 192)$ resolution is provided solely for visual comparison against an image with minimal compression artifacts but not further evaluated.

### A.1.2 Game Characteristics

As discussed by Trumpp et al. (2025), there are four Procgen environments (*Coinrun, Jumper, Ninja,* and *Caveflyer*) where the map in the background is not fixed but translated in x- and y-direction relatively with the agent, i.e., the agent remains in the center of the observation, but may look left/right or rotate.

### A.2 Hyperparameters List

Table 1: Hyperparameters for Proximal Policy Optimization (PPO).

| Hyperparameter | Values |
|---|---|
| Number Parallel Environments | 64 |
| Environment Steps | 256 |
| Learning Rate ($\tau = 2$) | $3.5 \times 10^{-4}$ |
| Batch Size | 2048 |
| Epochs | 3 |
| Discount Factor $\gamma$ | 0.99 |
| GAE Lambda ($\lambda$) | 0.95 |
| Clip Range | 0.2 |
| Value Function Coefficient | 0.5 |
| Entropy Coefficient | 0.01 |
| Max Gradient Norm | 0.5 |
| Optimizer | Adam |
| Shared Policy and Value Network | Yes |

## B  Extended Empirical Results

### B.1  Further Baselines

### B.1.1  Soft Mixture-of-Experts

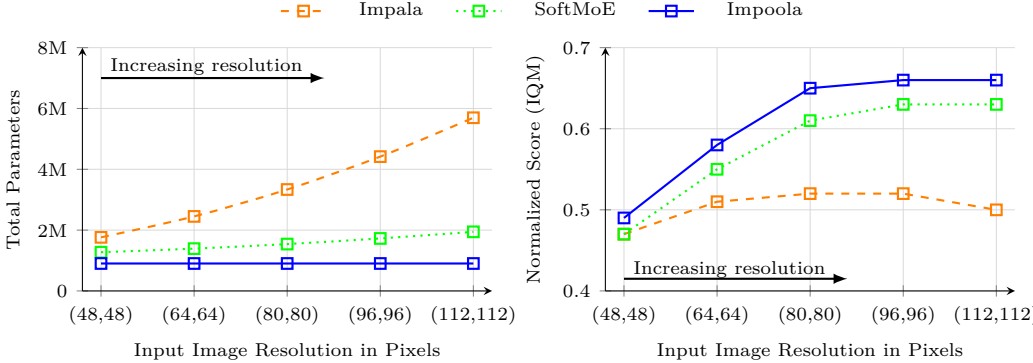

Figure 14: Environment-level comparison for all 16 Procgen-HD games for *generalization*, showing the final normalized scores for testing levels of Impala, Impoola, and additionally SoftMoE. All architectures use a width scale of $\tau = 3$; the resolution increases from $(64, 64)$ to $(112, 112)$. The counts include the parameters for separate actor and critic heads.

We evaluate the use of soft mixture-of-experts (SoftMoE) as proposed by Sokar et al. (2025) in comparison to the Impala and Impoola models. This soft mixture-of-experts (SoftMoE) uses the same backend as Impala, but instead of flattening the feature maps, they are tokenized and distributed to expert heads. Our results are based on the repository from Trumpp et al. (2025) (`https://github.com/raphajaner/impoola`), a PyTorch

reimplementation of the official code. We use 10 expert heads and follow the standard parameters with the PerConv tokenization otherwise. As shown in Figure 14, SoftMoE scales to higher resolutions substantially better than the standard Impala baseline. However, SoftMoE's routing and slotting mechanism causes its parameter count to still increase with resolution; this growth is less pronounced than Impala's rapid scaling, though. Ultimately, Impoola not only achieves superior overall performance, but it does so with a strictly resolution-independent parameter count and a significantly simpler network architecture and corresponding code implementation. This result supports our discussion that architectures that handle spatial features in a structured way can leverage additional visual detail, whereas Impala's flattening results in inefficient feature processing. As such, these results consistently extend the findings in Trumpp et al. (2025); Sokar & Castro (2025) that show the advantageous use of GAP, but also the potential of SoftMoEs.

### B.1.2 Further Variations of Impala Architecture

Increasing the image resolution (R,R) has two distinct effects on the base Impala architecture:

1. The parameter count of the projection layer weight matrix $W_{\text{proj}}$ increases quadratically with R, see the parameter counts in Figure 1 (left) and the per-layer architecture overview in Appendix C.

2. The absolute receptive field of the network, which is $r_{\text{field}} = 141$, covers the image range by $\approx 220\%$ for R=64, which decreases to $\approx 125\%$ for R=112.

We evaluate several variations of Impala that explicitly address these two compounding issues by introducing the following modifications to the base architecture:

- **Impala w/ Downsampling:** This structural modification applies aggressive striding early in the visual backbone, i.e., a kernel size of 4 for the first `Conv2d` layer in `ConvSeq_0` and a downsampling max-pooling with increased $s = 3$ stride and a kernel size of 4; the rest of the architecture remains the same. This modification increases the downsampling factor to $k = 12$, i.e., the feature map dimensions for R=(96,96) become $C = W = 8$, which the same as for the base configuration with R=(64,64). Moreover, this operation expands the absolute receptive field to $r_{\text{field}} = 204$, preserving a similar coverage ratio of $\approx 213\%$ for R=(96,96).

- **Impala w/ $d(z) = 50$ (Bottleneck):** Since the `Linear` projection layer's parameter count depends quadratically on the resolution R, this configuration decreases the output of the `Linear` projection from $d(z) = 256$ to $d(z) = 50$, thus introducing a stronger bottleneck while leading to a slower increase of the parameter count with R; the value of $d(z) = 50$ is motivated by the choice for the bottleneck layer proposed for DrQ-v2 (Yarats et al., 2022).

- **Impala w/ DrQv2-Style (Bottleneck):** This ablation builds upon the variation with the reduced bottleneck size of $d(z) = 50$ but also introduces a subsequent `LayerNorm` layer and `Tanh` activation to mimic the exact style of the bottleneck layer proposed by Yarats et al. (2022) for DrQ-v2.

- **Impala w/ 4×ConvSeq:** This variant appends a fourth hierarchical `ConvSeq_3` block to the backbone network. As calculated in Appendix B.4, this structural expansion scales the absolute receptive field to $r_{\text{field}} = 301$, allowing for a high image cover rates of $\approx 313\%$ at R=(96,96). However, at lower resolutions like $R = 64$, it may oversaturate the spatial context and collapse the final feature map to a restrictive $4 \times 4$ grid footprint with $\approx 470\%$ coverage ratio.

- **Impala w/ Kernel (5,5):** This variant increases the kernel size of all `Conv2d` blocks from the standard (3,3) to (5,5), which leads to an increased absolute receptive field of 281, covering the image to $\approx 292\%$ at R=96 but keeps the size of the `Linear` projection layer unchanged.

We present the results in Figure 15a and analyze the trends regarding how each structural variation responds to scaling the input resolution from $R = 64$ to $R = 96$ in the following.

First, it can be seen that increasing the absolute receptive field naively via a larger kernel size (5,5) across all layers is detrimental at the standard resolution and provides no benefit at the higher resolution, but recovers

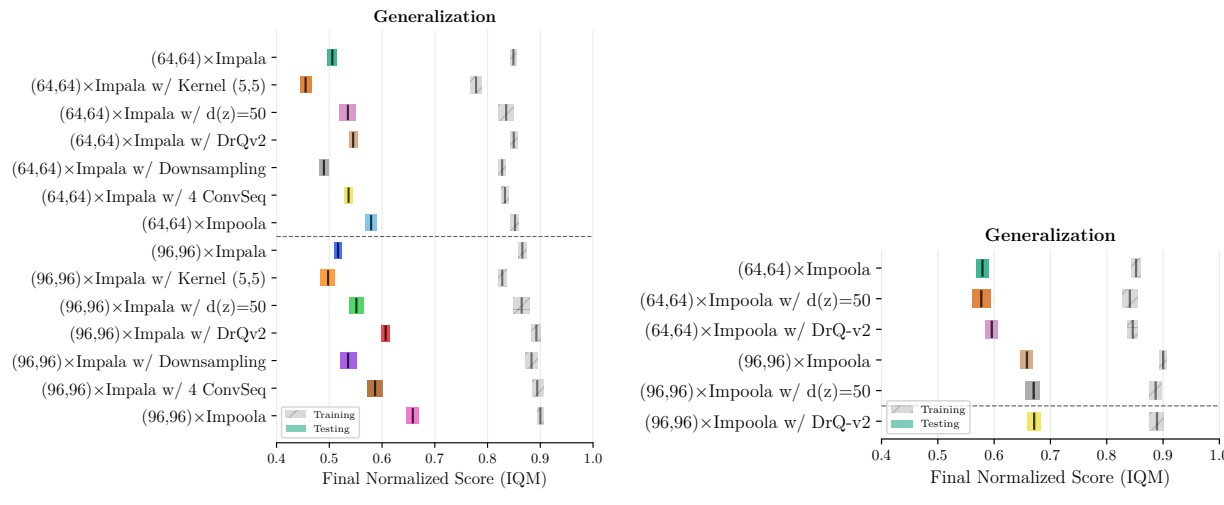

(a) Results for Impala variants.

(b) Results for reduced bottleneck in Impoola.

Figure 15: Aggregate results for the easy generalization track for variations of the base architectures. Results are shown for a width scale of $\tau = 3$, comparing performance when the resolution is scaled to (96,96).

performance to some extent. In contrast, focused downsampling in the first `ConvSeq` allows the model to better leverage the increased visual fidelity than the baseline. Nevertheless, these gains are rather modest compared with the gains from increasing network depth or introducing a bottleneck layer.

A notable trend emerges with the $d(z) = 50$ bottleneck configuration: introducing a stricter information constraint lowers training scores at both resolutions but consistently yields superior test performance compared to the Impala baseline. As such, compressing the latent space seems to act as a robust regularizer, which we assume is also, at least in parts, driving Impoola's performance. Performance can be further improved when mimicking the DrQ-v2 bottleneck layer that uses `LayerNorm`, which indicates that a structurally intact information bottleneck is a key property required to leverage increasing image resolutions.

In particular, the results for the deeper network, which performs better than the other modification, raise the question of which aspect is driving the improvement, since this modification increases the absolute receptive field while reducing the parameter count in the `Linear` projection layer due to the higher downsampling factor $k = 16$. We suspect that indeed both aspects work in parallel, i.e., the reduced parameter count in the `Linear` layer allows the network to better leverage the increased receptive field coverage.

Overall, while these results show that performance gains from visual scaling can also be found by other modifications of Impala's base architecture, it shows that Impoola's design using a `GAP` layer is preferable since it achieves the highest aggregated performance, alongside a simple implementation and a resolution-independent parameter count.

In addition, we also tested the implication of reducing the bottleneck layer in Impoola from $d(z) = 256$ to $d(z) = 50$, which results in an even more substantially reduced parameter count in the `Linear` projection layer. The results in Figure 15b show that this chance leads to a reduced performance on training levels. However, for the high-resolution image with R=96, we find that this yields slightly better testing performance. These results are consistent with our findings above, demonstrating that a further reduced bottleneck layer serves a regularizing function. As such, we reason that Impoola does not necessarily need further modifications to improve learning dynamics, since the `GAP` layer addresses this aspect; at the same time, it may not be detrimental, which may motivate the use of a `GAP` layer in other architectures as well.

### B.1.3 Per-Layer Dormant Neuron Analysis

In addition to the total number of dormant neurons in Figure 9, we provide a per-layer analysis of dormant neurons in the following. We show in Figure 16 six distinct layers, the very first `Conv2d` in the network ($l_0$),

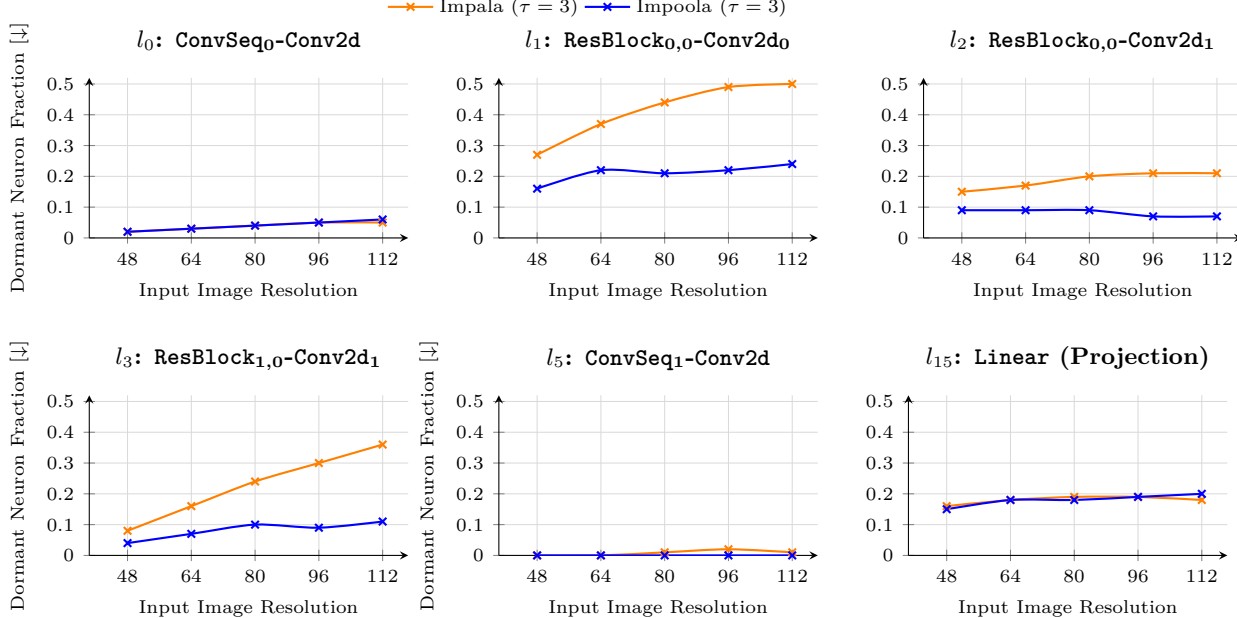

Figure 16: Layer-wise dormant neuron analysis comparing standard Impala and Impoola configurations at scale factor $\tau = 3$. It can be observed that dormant neurons accumulate in the deepest network layers $l_1$ to $l_4$ without residual connections to the next block in Impala, a pattern not observed in the Impoola architecture.

both `Conv2d` layers in the first `ResBlock`$_{0,0}$ ($l_1$ and $l_2$, respectively) and the first `Conv2d` ($l_3$) in the next `ResBlock`$_{0,1}$. Additionally, the next `ConvSeq`$_1$'s `Conv2d` ($l_5$) is shown alongside the `Linear` projection layer ($l_{15}$) of the backbone. See Appendix C for an overview of the network with per-layer details.

Our layer-wise analysis reveals that in the first layer of the network, $l_0$, and in layers after $l_5$, the overall count of dormant neurons is low, with only a modest increase with respect to resolution R.

However, we find evidence that dormant neurons accumulate particularly in Impala's deepest layers, namely $l_1$ rather than $l_0$, since $l_0$ is connected to earlier layers via residual connections. Similarly, $l_3$, the other first layer in the next `ResBlock` exhibits a similar behavior. While the overall count is lower than in $l_0$, there is still an evident correlation to the input resolution R.

As such, this result suggests that for the Impala architecture, the first `ConSeq`$_0$ outputs almost an identity mapping by using the shortcut path of the residual connection to the output, i.e., the input may not be substantially altered by the `Con2d` layers in these two `ResBlock`.

This finding points to a structural interplay between the spatial parameter explosion of the standard Impala approach using a `Flatten` layer and the degradation of the deeper `Conv2d` layers in the backbone. We suspect that related parameter explosion in the `Linear` layer of Impala may cause gradient issues in these deep network layers ($l_1$ to $l_4$), whereas the network learns to not actively use the parameters in these layers but instead falls back to the residual connection to the next `ConvSeq`$_1$. In contrast, Impoola shows only a modest increase of dormant neurons in these deep layers, suggesting that the structural bottleneck introduced by the `GAP` layer ensures that the network leverages its full depth and is not falling back to the residual connections.

It is noteworthy that while the relative dormant neuron count in the `Linear` layer ($l_{15}$) is about the same as in Impoola, it means that the absolute count is still substantially higher due to the disproportionate parameter allocation to it in Impala, see Appendix C.

## B.2 Per Environment Resolution Scaling

### B.2.1 Generalization

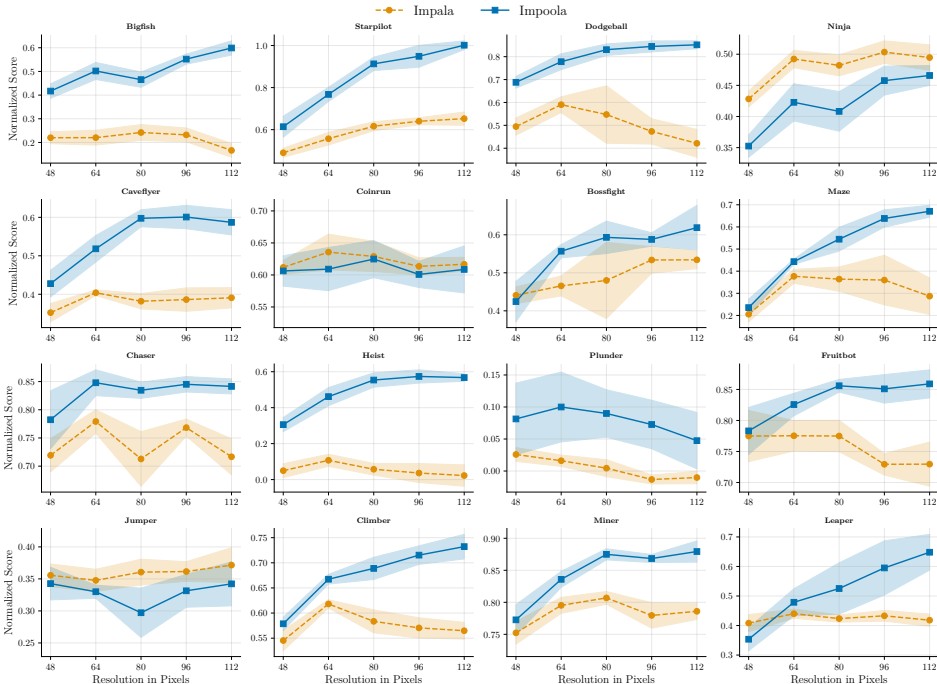

Figure 17: Environment-level comparison for all 16 Procgen-HD games for generalization, showing the final normalized scores of Impala and Impoola for testing levels. Both architectures use a width scale of $\tau = 2$; the resolution increases from $(48, 48)$ to $(112, 112)$.

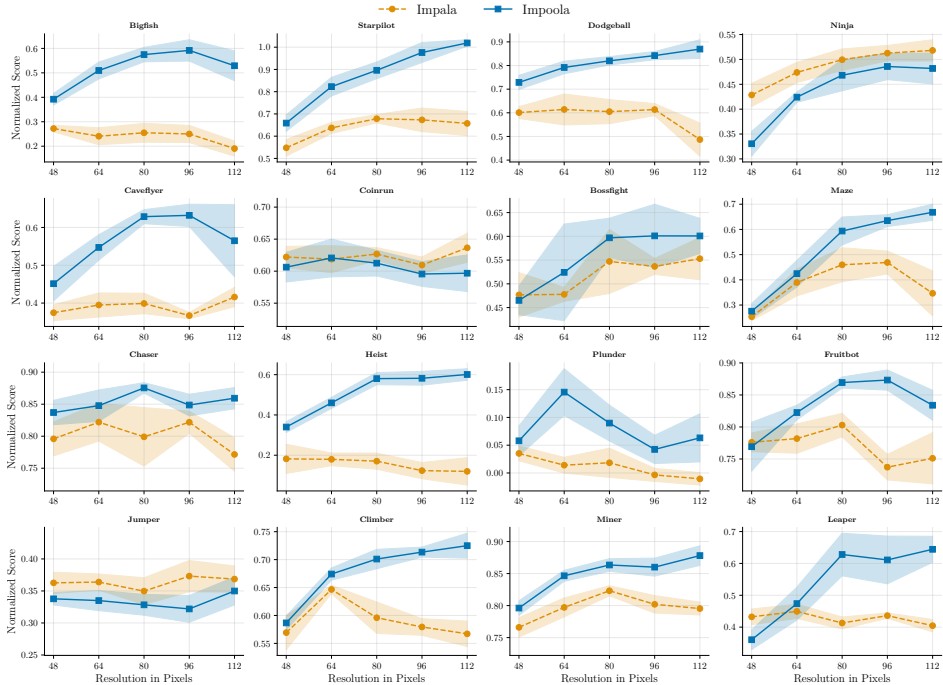

Figure 18: Environment-level comparison for all 16 Procgen-HD games for generalization, showing the final normalized scores of Impala and Impoola for testing levels. Both architectures use a width scale of $\tau = 3$; the resolution increases from $(48, 48)$ to $(112, 112)$.

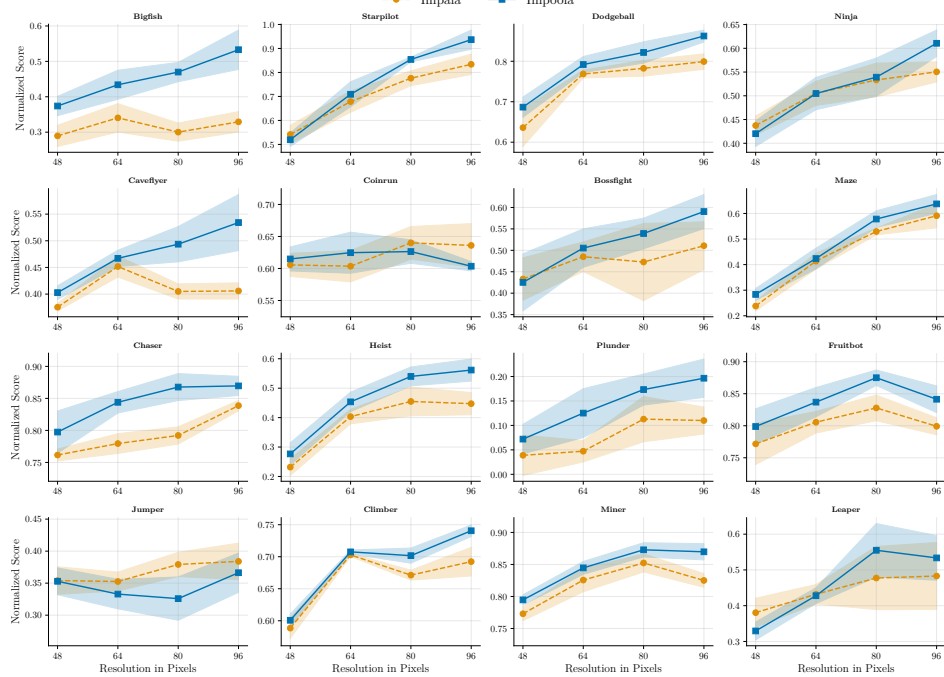

Figure 19: Environment-level comparison for all 16 Procgen-HD games for generalization, showing the final normalized scores of Impala and Impoola for testing levels. Both architectures use a *deeper* architecture with four `ConvSeq` elements and a width scale of $\tau = 3$; the resolution increases from $(48, 48)$ to $(112, 112)$.

## B.2.2 Hard Generalization

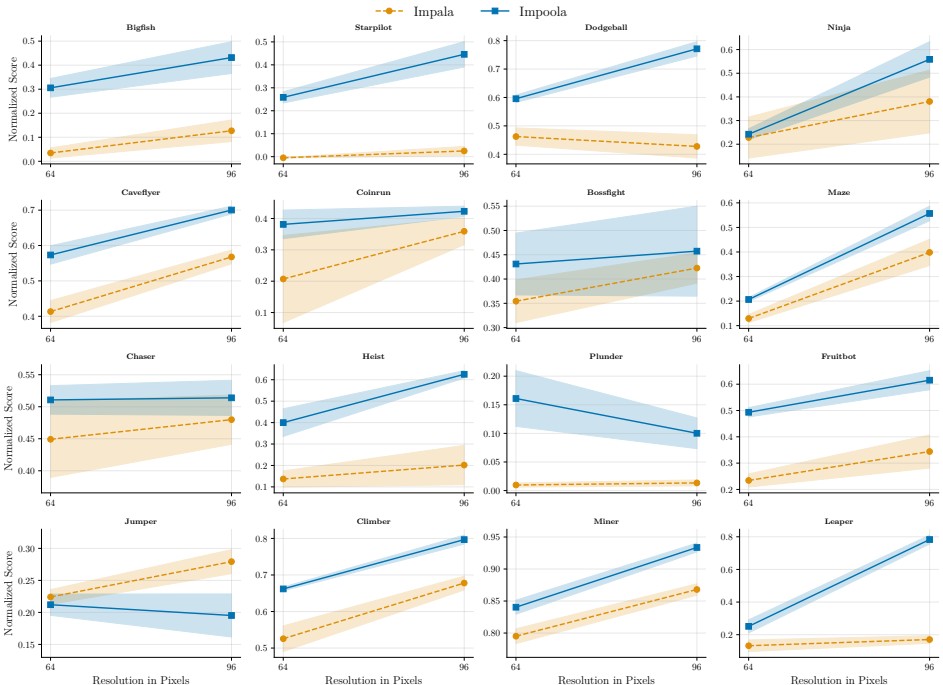

Figure 20: Environment-level comparison for all 16 Procgen-HD games for *hard* generalization, showing the final normalized scores of Impala and Impoola for testing levels. Both architectures use a width scale of $\tau = 2$; the resolution increases from $(64, 64)$ to $(96, 96)$.

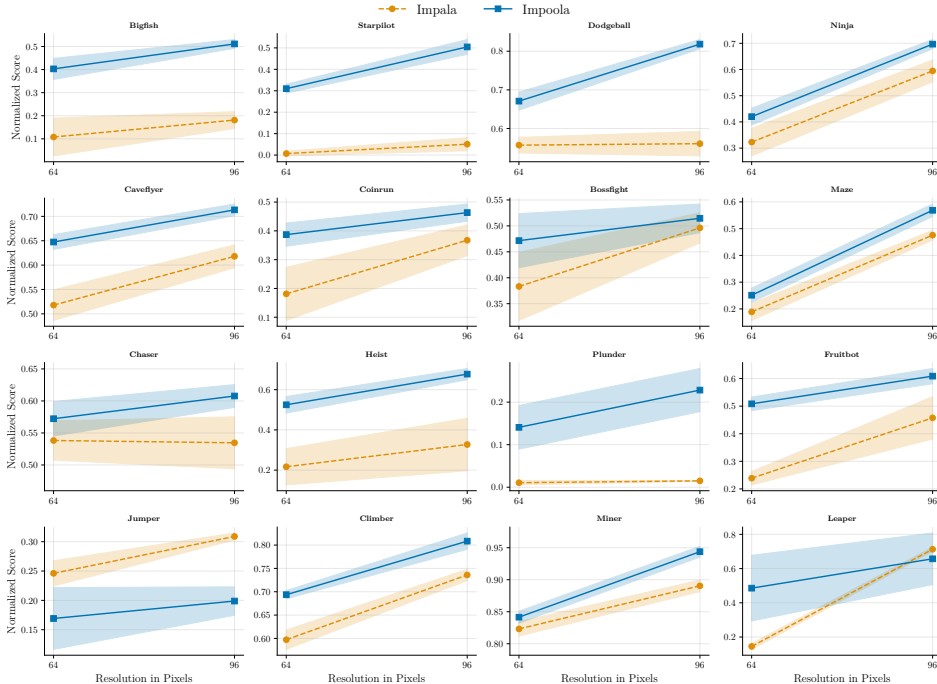

Figure 21: Environment-level comparison for all 16 Procgen-HD games for *hard* generalization, showing the final normalized scores of Impala and Impoola for testing levels. Both architectures use a width scale of $\tau = 3$; the resolution increases from $(64, 64)$ to $(96, 96)$.

### B.2.3 Efficiency

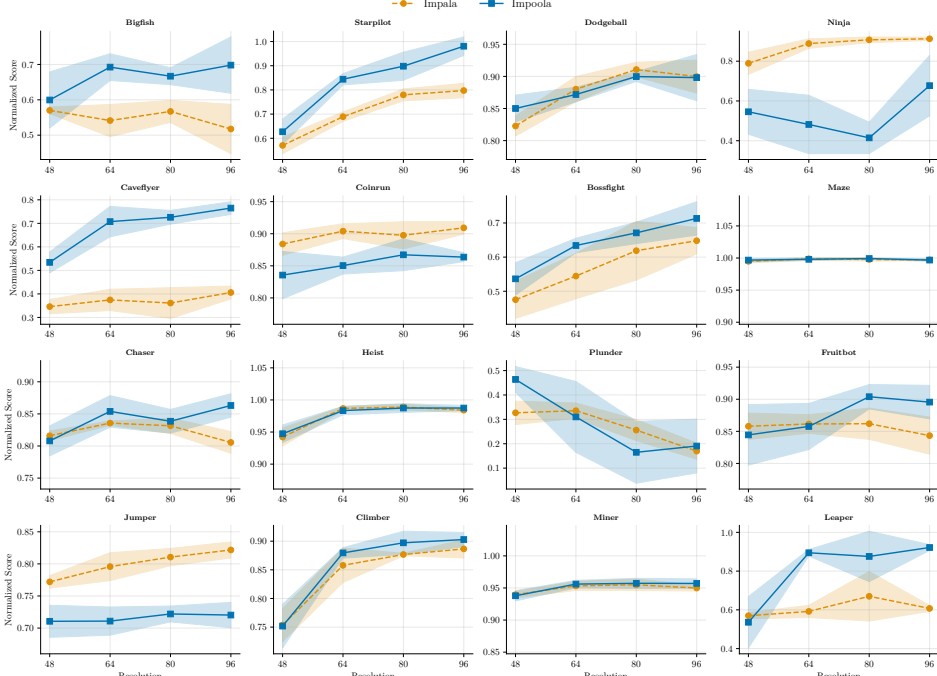

Figure 22: Environment-level comparison for all 16 Procgen-HD games for *efficiency*, showing the final normalized scores of Impala and Impoola for testing levels. Both architectures use a width scale of $\tau = 3$; the resolution increases from $(48, 48)$ to $(112, 112)$.

### B.3 Computational Cost

**Note:** Presented times are approximate measurements using Python timers, averaged over 3 seeds for the Starpilot game, on a server with an AMD EPYC 7763 64-Core Processor (2P) CPU and an NVIDIA A100 PCIe 40GB GPU. Our code is based on PyTorch, but no `torch.compile()` is used. In each iteration, the *full* batch of recorded trajectories is loaded into VRAM, and mini-batch sizes according to Table A.2 are used. We use `torch.cuda.max_memory_allocated()` to measure VRAM usage.

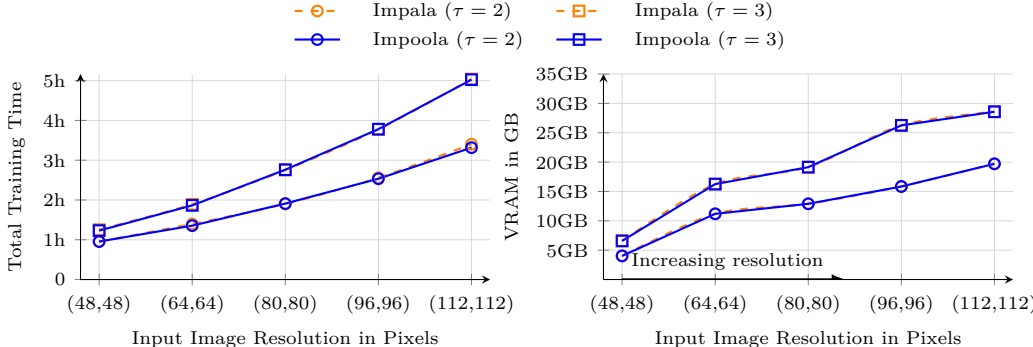

Figure 23: The impact of scaling the input image resolution on the total training time. We compare the common image encoders Impala and Impoola across resolutions from $(48, 48)$ to $(112, 112)$ pixels. Different network widths are shown, i.e., the number of filters per Conv2d layer is scaled by $\tau$.

Figure 23 illustrates the impact of input resolution and network width scale $\tau$ on total training time and VRAM allocation. We find that Impoola and Impala exhibit almost identical computational costs across all tested configurations. Replacing the standard flattening operation with GAP does not provide a measurable benefit in either total training time or memory footprint. As such, the computational demand is driven entirely by the shared convolutional backbone and scales rapidly with the input resolution. For instance, increasing the resolution from $(64, 64)$ to $(112, 112)$ for a wider network with $\tau = 3$ inflates the training time from less than 2 h to approximately 5h, while VRAM consumption grows from roughly 16 GB to nearly 30 GB. We find the VRAM increase to be non-monotonic, e.g., exhibiting a plateau at the $(80, 80)$ resolution. We reason that this step-wise allocation behavior likely stems from dynamic hardware-level heuristics, such as cuDNN algorithm selection and cache size adjustments. In summary, it can be seen that the performance and generalization improvements from increased image resolutions can only be achieved when sufficient compute is available.

Furthermore, we also present which parts of the training contribute the most to the increase, splitting the training time for Impala and Impoola with $\tau = 3$ into 5 distinct steps as shown in Figure 24. It can be seen that the backward step for gradient calculation grows strongly with higher resolutions; thus, we see significant potential for speed improvements by optimizing the PyTorch implementation in future work.

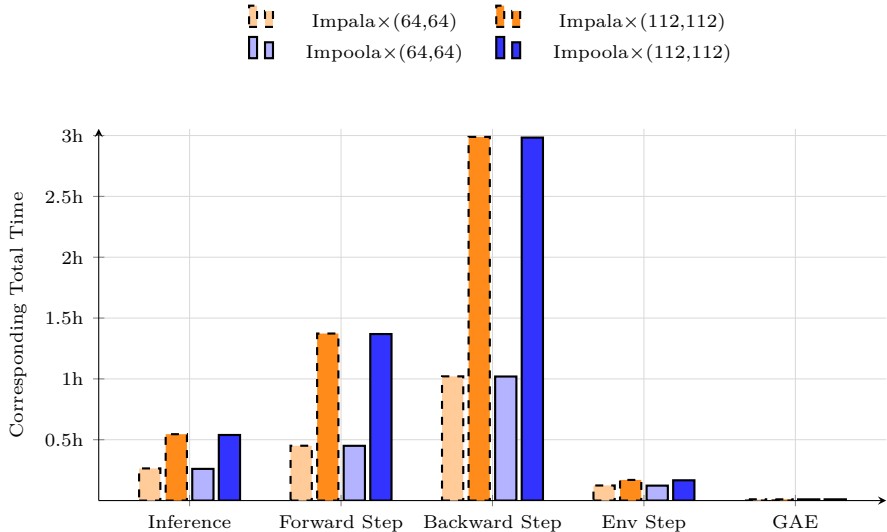

Figure 24: Breakdown of total training time, comparing the duration of the inference step (`a=model(x)`), environment step (`x'=env(a)`), GAE calculation, forward step (**loss=...**), and the backward step (`loss.backward()` calculation) for Impoola and Impala at $(64, 64)$ and $(112, 112)$ using $\tau = 3$. Other small steps included in the total training time, e.g., logging, is not depicted.

## B.4 Receptive Field Analysis

The theoretical absolute receptive field defines the physical window size in the original input pixels that a single feature activation of the output feature map $m$ can observe, independent of the total input resolution $R$. As such, it allows measuring the fraction of the input image that a single feature activation in $m$ *can see.*

We calculate the absolute receptive field size $r_l$ and the cumulative spatial stride $j_l$ at any given layer $l$ by tracking recursively from the input layer forward using the standard formulation

$$j_l = j_{l-1} \cdot s_l, \tag{3}$$

$$r_l = r_{l-1} + (k_l - 1) \cdot j_{l-1}, \tag{4}$$

where $k_l$ represents the kernel size of the operation at layer $l$, and $s_l$ denotes its stride. The base initialization at the raw input image ($l = 0$) is defined as $r_0 = 1$ and $j_0 = 1$.

For the base architecture of Impala, the calculation is as follows:

1. **ConvSeq$_0$:**
   - $l_0$ (ConvSeq$_0$-Conv2d): $k = 3, s = 1 \implies j_1 = 1 \cdot 1 = 1 \implies r_1 = 1 + (3 - 1) \cdot 1 = 3$
   - $l_{0'}$ (ConvSeq$_0$-MaxPool2d): $k = 3, s = 2 \implies j_2 = 1 \cdot 2 = 2 \implies r_2 = 3 + (3 - 1) \cdot 1 = 5$
   - $l_1$ (ResBlock$_{0,0}$-Conv2d): $k = 3, s = 1 \implies j_3 = 2 \cdot 1 = 2 \implies r_3 = 5 + (3 - 1) \cdot 2 = 9$
   - $l_2$ (ResBlock$_{0,0}$-Conv2d): $k = 3, s = 1 \implies j_4 = 2 \cdot 1 = 2 \implies r_4 = 9 + (3 - 1) \cdot 2 = 13$
   - $l_3$ (ResBlock$_{0,1}$-Conv2d): $k = 3, s = 1 \implies j_5 = 2 \cdot 1 = 2 \implies r_5 = 13 + (3 - 1) \cdot 2 = 17$
   - $l_4$ (ResBlock$_{0,1}$-Conv2d): $k = 3, s = 1 \implies j_6 = 2 \cdot 1 = 2 \implies r_6 = 17 + (3 - 1) \cdot 2 = 21$

2. **ConvSeq$_1$:**
   - $l_5$ (ConvSeq$_1$-Conv2d): $k = 3, s = 1 \implies j_7 = 2 \cdot 1 = 2 \implies r_7 = 21 + (3 - 1) \cdot 2 = 25$
   - $l_{5'}$ (ConvSeq$_1$-MaxPool2d): $k = 3, s = 2 \implies j_8 = 2 \cdot 2 = 4 \implies r_8 = 25 + (3 - 1) \cdot 2 = 29$
   - $l_6$ (ResBlock$_{1,0}$-Conv2d): $k = 3, s = 1 \implies j_9 = 4 \cdot 1 = 4 \implies r_9 = 29 + (3 - 1) \cdot 4 = 37$
   - $l_7$ (ResBlock$_{1,0}$-Conv2d): $k = 3, s = 1 \implies j_{10} = 4 \cdot 1 = 4 \implies r_{10} = 37 + (3 - 1) \cdot 4 = 45$
   - $l_8$ (ResBlock$_{1,1}$-Conv2d): $k = 3, s = 1 \implies j_{11} = 4 \cdot 1 = 4 \implies r_{11} = 45 + (3 - 1) \cdot 4 = 53$
   - $l_9$ (ResBlock$_{1,1}$-Conv2d): $k = 3, s = 1 \implies j_{12} = 4 \cdot 1 = 4 \implies r_{12} = 53 + (3 - 1) \cdot 4 = 61$

3. **ConvSeq$_2$:**
   - $l_{10}$ (ConvSeq$_2$-Conv2d): $k = 3, s = 1 \implies j_{13} = 4 \cdot 1 = 4 \implies r_{13} = 61 + (3 - 1) \cdot 4 = 69$
   - $l_{10'}$ (ConvSeq$_2$-MaxPool2d): $k = 3, s = 2 \implies j_{14} = 4 \cdot 2 = 8 \implies r_{14} = 69 + (3 - 1) \cdot 4 = 77$
   - $l_{11}$ (ResBlock$_{2,0}$-Conv2d): $k = 3, s = 1 \implies j_{15} = 8 \cdot 1 = 8 \implies r_{15} = 77 + (3 - 1) \cdot 8 = 93$
   - $l_{12}$ (ResBlock$_{2,0}$-Conv2d): $k = 3, s = 1 \implies j_{16} = 8 \cdot 1 = 8 \implies r_{16} = 93 + (3 - 1) \cdot 8 = 109$
   - $l_{13}$ (ResBlock$_{2,1}$-Conv2d): $k = 3, s = 1 \implies j_{17} = 8 \cdot 1 = 8 \implies r_{17} = 109 + (3 - 1) \cdot 8 = 125$
   - $l_{14}$ (ResBlock$_{2,1}$-Conv2d): $k = 3, s = 1 \implies j_{18} = 8 \cdot 1 = 8 \implies r_{18} = 125 + (3 - 1) \cdot 8 = \mathbf{141}$

We omit the detailed calculations for the variation of the Impala architecture listed in Appendix B.1.2 for brevity and summarize the downsampling factors, absolute receptive field values, and corresponding image coverage rates directly in Table 2.

Surprisingly, the absolute receptive field of the base Impala with $r_{\text{field}} = 141$ is large enough to fully cover the input image, even at R=112. Note that Impoola and Impala w/ Bottleneck ($\mathrm{d}(z) = 50$ and DrQ-v2 style) have the same downsampling metrics and absolute receptive field as the base Impala architecture.

Table 2: Downsampling factor $k$, absolute receptive fields ($r_{\text{field}}$) for $l_{14}$, feature map footprints ($H \times W$), and relative image coverage percentages across input dimensions ($R$).

| Variant | $k$ | $r_{\text{field}}$ | $R = 64$ | | $R = 96$ | | $R = 112$ | |
| | | | Grid | Coverage | Grid | Coverage | Grid | Coverage |
|---|---|---|---|---|---|---|---|---|
| Impala | 8 | 141 | $8 \times 8$ | $\approx 220\%$ | $12 \times 12$ | $\approx 147\%$ | $14 \times 14$ | $\approx 125\%$ |
| Impala w/ $4\times$ConvSeq | 16 | 301 | $4 \times 4$ | $\approx 470\%$ | $6 \times 6$ | $\approx 313\%$ | $7 \times 7$ | $\approx 268\%$ |
| Impala w/ Kernel (5,5) | 8 | 281 | $8 \times 8$ | $\approx 439\%$ | $12 \times 12$ | $\approx 292\%$ | $14 \times 14$ | $\approx 250\%$ |
| Impala w/ Downsampling | 12 | 204 | $5 \times 5$ | $\approx 319\%$ | $8 \times 8$ | $\approx 213\%$ | $9 \times 9$ | $\approx 182\%$ |

# C  Per-Layer Network Architecture

Table 3: Model summary of the Impala-CNN ($\tau = 3$) for PPO with 96 x 96 input images. The overall parameter count is 4,416,720, with a total of 590.23M multi-adds.

| Layer (type:depth-idx) | Input | Output | Param # | Kernel | Param % | Multi-Adds |
|---|---|---|---|---|---|---|
| ImpalaPPOActorCritic | [3, 96, 96] | [15] | – | – | – | – |
| Impala-CNN | [3, 96, 96] | [256] | – | – | – | – |
| ConvSequence | [3, 96, 96] | [48, 48, 48] | – | – | – | – |
| $l_0$: Conv2d | [3, 96, 96] | [48, 96, 96] | 1,344 | [3, 3] | 0.03% | 12,386,304 |
| ResidualBlock | [48, 48, 48] | [48, 48, 48] | – | – | – | – |
| $l_1$: Conv2d | [48, 48, 48] | [48, 48, 48] | 20,784 | [3, 3] | 0.47% | 47,886,336 |
| $l_2$: Conv2d | [48, 48, 48] | [48, 48, 48] | 20,784 | [3, 3] | 0.47% | 47,886,336 |
| ResidualBlock | [48, 48, 48] | [48, 48, 48] | – | – | – | – |
| $l_3$: Conv2d | [48, 48, 48] | [48, 48, 48] | 20,784 | [3, 3] | 0.47% | 47,886,336 |
| $l_4$: Conv2d | [48, 48, 48] | [48, 48, 48] | 20,784 | [3, 3] | 0.47% | 47,886,336 |
| ConvSequence | [48, 48, 48] | [96, 24, 24] | – | – | – | – |
| $l_5$: Conv2d | [48, 48, 48] | [96, 48, 48] | 41,568 | [3, 3] | 0.94% | 95,772,672 |
| ResidualBlock | [96, 24, 24] | [96, 24, 24] | – | – | – | – |
| $l_6$: Conv2d | [96, 24, 24] | [96, 24, 24] | 83,040 | [3, 3] | 1.88% | 47,831,040 |
| $l_7$: Conv2d | [96, 24, 24] | [96, 24, 24] | 83,040 | [3, 3] | 1.88% | 47,831,040 |
| ResidualBlock | [96, 24, 24] | [96, 24, 24] | – | – | – | – |
| $l_8$: Conv2d | [96, 24, 24] | [96, 24, 24] | 83,040 | [3, 3] | 1.88% | 47,831,040 |
| $l_9$: Conv2d | [96, 24, 24] | [96, 24, 24] | 83,040 | [3, 3] | 1.88% | 47,831,040 |
| ConvSequence | [96, 24, 24] | [96, 12, 12] | – | – | – | – |
| $l_{10}$: Conv2d | [96, 24, 24] | [96, 24, 24] | 83,040 | [3, 3] | 1.88% | 47,831,040 |
| ResidualBlock | [96, 12, 12] | [96, 12, 12] | – | – | – | – |
| $l_{11}$: Conv2d | [96, 12, 12] | [96, 12, 12] | 83,040 | [3, 3] | 1.88% | 11,957,760 |
| $l_{12}$: Conv2d | [96, 12, 12] | [96, 12, 12] | 83,040 | [3, 3] | 1.88% | 11,957,760 |
| ResidualBlock | [96, 12, 12] | [96, 12, 12] | – | – | – | – |
| $l_{13}$: Conv2d | [96, 12, 12] | [96, 12, 12] | 83,040 | [3, 3] | 1.88% | 11,957,760 |
| $l_{14}$: Conv2d | [96, 12, 12] | [96, 12, 12] | 83,040 | [3, 3] | 1.88% | 11,957,760 |
| Flatten | [96, 12, 12] | [13824] | – | – | – | – |
| $l_{15}$: Linear | [13824] | [256] | 3,539,200 | – | 80.13% | 3,539,200 |
| Actor | [256] | [15] | 3,855 | – | 0.09% | 3,855 |
| Critic | [256] | [1] | 257 | – | 0.01% | 257 |

Table 4: Model summary of the Impoola-CNN ($\tau = 3$) for PPO with 96 x 96 input images. The overall parameter count is 902,352, with a total of 586.72M multi-adds.

| Layer (type:depth-idx) | Input | Output | Param # | Kernel | Param % | Multi-Adds |
|---|---|---|---|---|---|---|
| ImpoolaPPOActorCritic | [3, 96, 96] | [15] | – | – | – | – |
| Impoola-CNN | [3, 96, 96] | [256] | – | – | – | – |
| ConvSequence | [3, 96, 96] | [48, 48, 48] | – | – | – | – |
| $l_0$: Conv2d | [3, 96, 96] | [48, 96, 96] | 1,344 | [3, 3] | 0.15% | 12,386,304 |
| ResidualBlock | [48, 48, 48] | [48, 48, 48] | – | – | – | – |
| $l_1$: Conv2d | [48, 48, 48] | [48, 48, 48] | 20,784 | [3, 3] | 2.30% | 47,886,336 |
| $l_2$: Conv2d | [48, 48, 48] | [48, 48, 48] | 20,784 | [3, 3] | 2.30% | 47,886,336 |
| ResidualBlock | [48, 48, 48] | [48, 48, 48] | – | – | – | – |
| $l_3$: Conv2d | [48, 48, 48] | [48, 48, 48] | 20,784 | [3, 3] | 2.30% | 47,886,336 |
| $l_4$: Conv2d | [48, 48, 48] | [48, 48, 48] | 20,784 | [3, 3] | 2.30% | 47,886,336 |
| ConvSequence | [48, 48, 48] | [96, 24, 24] | – | – | – | – |
| $l_5$: Conv2d | [48, 48, 48] | [96, 48, 48] | 41,568 | [3, 3] | 4.61% | 95,772,672 |
| ResidualBlock | [96, 24, 24] | [96, 24, 24] | – | – | – | – |
| $l_6$: Conv2d | [96, 24, 24] | [96, 24, 24] | 83,040 | [3, 3] | 9.20% | 47,831,040 |
| $l_7$: Conv2d | [96, 24, 24] | [96, 24, 24] | 83,040 | [3, 3] | 9.20% | 47,831,040 |
| ResidualBlock | [96, 24, 24] | [96, 24, 24] | – | – | – | – |
| $l_8$: Conv2d | [96, 24, 24] | [96, 24, 24] | 83,040 | [3, 3] | 9.20% | 47,831,040 |
| $l_9$: Conv2d | [96, 24, 24] | [96, 24, 24] | 83,040 | [3, 3] | 9.20% | 47,831,040 |
| ConvSequence | [96, 24, 24] | [96, 12, 12] | – | – | – | – |
| $l_{10}$: Conv2d | [96, 24, 24] | [96, 24, 24] | 83,040 | [3, 3] | 9.20% | 47,831,040 |
| ResidualBlock | [96, 12, 12] | [96, 12, 12] | – | – | – | – |
| $l_{11}$: Conv2d | [96, 12, 12] | [96, 12, 12] | 83,040 | [3, 3] | 9.20% | 11,957,760 |
| $l_{12}$: Conv2d | [96, 12, 12] | [96, 12, 12] | 83,040 | [3, 3] | 9.20% | 11,957,760 |
| ResidualBlock | [96, 12, 12] | [96, 12, 12] | – | – | – | – |
| $l_{13}$: Conv2d | [96, 12, 12] | [96, 12, 12] | 83,040 | [3, 3] | 9.20% | 11,957,760 |
| $l_{14}$: Conv2d | [96, 12, 12] | [96, 12, 12] | 83,040 | [3, 3] | 9.20% | 11,957,760 |
| AdaptiveAvgPool2d | [96, 12, 12] | [96, 1, 1] | – | – | – | – |
| Flatten | [96, 1, 1] | [96] | – | – | – | – |
| $l_{15}$: Linear | [96] | [256] | 24,832 | – | 2.75% | 24,832 |
| Actor | [256] | [15] | 3,855 | – | 0.43% | 3,855 |
| Critic | [256] | [1] | 257 | – | 0.03% | 257 |

