# OpenReview forum: "Higher Resolution, Better Generalization: Unlocking Visual Scaling in Deep Reinforcement Learning"
_TMLR — Under review for TMLR_

### Review · Reviewer_6cjc · 2026-03-23

**Summary Of Contributions:**

The paper's primary contributions revolve around breaking the "low-resolution paradigm" that has dominated the field since the early days of Atari benchmarks.
- Resolution as a First-Order Variable: The authors demonstrate that higher-resolution inputs can substantially improve both performance and generalization across various environments,
provided the network architecture can process them effectively.
- Identification of the "Flattening" Bottleneck: They reveal that the widely used Impala encoder fails to scale because its "Flatten" layer causes parameter counts to grow quadratically with
resolution, leading to a severe capacity allocation asymmetry.
- Validation of Resolution-Independent Architectures: By replacing flattening with Global Average Pooling (GAP) used in the Impoola architecture, the authors decouple parameter count from
input resolution. This shift allows the model to leverage higher visual detail, resulting in a 28% performance gain over Impala at their respective best conditions.
- Mechanistic Insights: Using gradient saliency and dormant neuron analysis, they show that higher resolution allows for more spatially localized attention on task-critical objects and helps
maintain active representations (fewer dormant neurons).
- Release of Procgen-HD: To facilitate future research, the authors introduced Procgen-HD, an open-source extension of the Procgen benchmark that supports arbitrary rendering resolutions
while keeping game logic and rewards identical.

Key Strengths:
- Simplicity and Practicality: The proposed solution (GAP/Impoola) is a straightforward architectural change that provides immediate benefits for visual RL tasks without complex algorithmic
tuning. The authors used resolution-independent architectures (such as ResNet) which are very well-known in computer vision domain.
- In-depth Analysis: Beyond just reporting higher scores, the paper explains why Impoola performs better than Impala and why higher resolution improves performance through representational
analysis, giving the readers a better understanding of the underlying reasons.

Key Limitations/Weaknesses:
- Scope Constraints: The study is focused on the Procgen benchmark (2D, discrete actions, PPO algorithm). It remains to be seen how these visual scaling laws apply to continuous control, 3D
environments, or off-policy methods.
- Absense of Explanation to Exceptional Observations: The authors claimed throughout the paper that Impoola outperforms Impala, but in some cases, the opposite has been observed in their
experiments. No explanation to these exceptional cases has been provided in the paper.

**Audience:**

Yes

**Audience Explanation:**

The paper addresses a foundational "hidden" convention in deep reinforcement learning (RL) and offers both a practical architectural solution and a new benchmarking tool.

The following sets of individuals in TMLR's audience will be interested to know the findings of this research:
1. Reinforcement Learning (RL) Researchers
- Performance Breakthroughs: The paper demonstrates that simply increasing observation resolution can unlock a 28% performance gain for agents without requiring any changes to the
underlying RL algorithms.
- Addressing Generalization: A core interest for TMLR readers is the "generalization gap". This paper shows that higher resolutions help narrow the gap between training and testing performance
by providing more visual detail.
- Mechanistic Insights: Researchers interested in why models fail or succeed would value the "mechanistic evidence" provided through dormant neuron analysis and gradient saliency maps,
which explain how higher resolution leads to more focused policy attention.

2. Deep Learning Architecture Designers
- Identifying Bottlenecks: The study highlights a critical flaw in the widely used Impala encoder - its "Flatten" layer causes parameter counts to grow quadratically with resolution, leading to
"capacity allocation asymmetry".
- Scalable Design: Designers would be interested in the Impoola architecture, which uses Global Average Pooling (GAP) to decouple parameter counts from input resolution. This allows the
model to scale effectively to high-fidelity tasks without a massive increase in model size.

3. Benchmark Designers and Evaluators
- New Research Tools: The introduction of Procgen-HD—an open-source extension that allows for arbitrary rendering resolutions while keeping game logic identical—provides a new tool for the
community to study visual scaling laws.
- Challenging Conventions: The paper critiques the "low-resolution paradigm" (e.g., $84 \times 84$ or $64 \times 64$ pixels) as an outdated standard born from 2013-era hardware limitations
rather than principled design.

4. Robotics and Computer Vision Practitioners
- Real-World Application: The robotics community frequently deals with high-resolution imagery where standard RL downsampling would discard critical information.
- Cross-Domain Learning: The paper bridges the gap between Computer Vision, where resolution is a tunable parameter, and RL, where it has traditionally been a rigid constraint.

**Claims And Evidence:**

Yes

**Claims Explanation:**

Some of the claimed made by the authors were supported by clear convincing evidence.

The authors claimed that higher-resolution inputs can substantially improve both performance and generalization across various environments, provided the network architecture can process
them effectively. Their experiments showed that across 16 Procgen environments, five resolutions, three network widths, and multiple training regimes, higher-resolution
inputs yield substantial improvement in aggregate performance and generalization without any algorithmic changes, but only when paired with a resolution-independent architecture. Performance
improvement was seen the most in those environments where precise perception of small or distant entities matters most, which clearly explains the importance of higher resolution.

The authors claimed that the widely used Impala encoder fails to scale because its "Flatten" layer causes parameter counts to grow quadratically with resolution, leading to a severe capacity
allocation asymmetry. Their experiments showed that Impala exhibits a clear monotonic increase in dormant neurons as resolution grows.

The authors claimed that when sufficient performance headroom remains, higher resolution reliably improves performance. For the hard generalization track, even Impala benefitted from higher
resolution because there was a performance headroom. However in efficiency track, there was very little a higher resolution could help, so the performance improvement was also small.

However, for a few other claims, the authors didn't provide explanation to the exceptional observations.

The authors claimed that in efficiency track (i.e. where the level restrictions are removed entirely, exposing agents to the full  procedural distribution during training), Impoola benefits from higher
resolution and outperforms Impala across all configurations. For $\tau$ = 3, 13 out of 16 environments support this claim, but in Ninja, Coinrun and Jumper environments, Impala consistently
outperformed Impoola.

For hard generalization track in Jumper environment, Impala consistently outperforms Impoola. This is an exception which doesn't match the prevalent tone of the paper.

**Requested Changes:**

The following adjustments are recommended to strengthen the work:
- In Figure 6, there is an anomaly - normalized score degraded for higher resolutions in Plunder environment. Figure 14, 15, 16 have some clues
(possible underfitting by smaller models), but the anomaly is not pointed out or explained anywhere. It is recommended that the anomaly is
properly addressed.
- For hard generalization track in Jumper environment, Impala consistently outperforms Impoola (as seen in Figure 17 and 18). Despite being an
exception, this observation is a direct contradiction to the claim presented in the paper. Explanation to this exception is recommended.
- In page 9, the authors claimed that "Impoola continues to benefit from higher resolution and outperforms Impala across all configurations", but
Figure 19 doesn't properly support this claim. In Ninja, Coinrun and Jumper environments, Impala consistently outperformed Impoola for
$\tau$ = 3. Explanation to this issue is strongly recommended.

---

> ### Author Response · Authors · 2026-04-10
> **Revised Manuscript**
>
> We thank Reviewer 6cjc for their thorough reading of our manuscript and their detailed feedback.
>
> Before discussing the individual points, we want to note that several of the Reviewer's observations regarding specific environments where Impala outperforms Impoola (Jumper, Coinrun, Ninja) are consistent with findings reported in the original Impoola paper (Trumpp et al., 2025), which identifies a strong inductive bias of Impala for agent-centered games. We have revised our paper to explicitly discuss these results in reference to Trumpp et al. (2025). Our general claims are framed at the aggregate level across all 16 environments, and we present per-environment results transparently so the reader can see where the general trend holds and where it does not.
>
> We incorporated the requested changes in the revised manuscript as follows:
>
> - **Point 1 - Plunder anomaly Figure 6:** We have added a discussion of this in Section 4.4 (now RQ4). Plunder as an environment is a general exemption since both architectures score very low, even during training (scores of approx. 0.15). At these performance levels, the agent has not yet learned a meaningful policy, so there is little for higher resolution to improve, if not to complicate things, as the agent has to deal with a larger feature space. We reason that the challenge in Plunder appears to be reward structure and exploration, not perception. In the hard generalization setting (Figures 18, 19), where more training levels are available, the trend for Plunder is more positive, consistent with our finding that resolution benefits appear when sufficient performance headroom exists.
>
> - **Point 2 - Impala/Impoola inconsistency for Jumper:** We have revised Section 4.4 (RQ4) to discuss this. As discussed in our first paragraph, Jumper is one of the agent-centered environments where Impala's inductive bias gives it an advantage. The (64,64) resolution already resolves the critical visual elements in this game, leaving little room for resolution scaling to help. While Impala wins here, the aggregate results across all 16 environments consistently favor Impoola.
>
> - **Point 3 - Impala/Impoola inconsistency for efficiency track:** We have corrected this in the revised manuscript: the text now states that Impoola outperforms Impala *on average across resolutions*, which accurately reflects the aggregate finding while acknowledging the per-environment exceptions. As noted in our first paragraph, these three environments (Ninja, Coinrun, and Jumper) fall into the same category of agent-centered games benefiting Impala. The revised Section 4.4 now explicitly discusses this pattern. We also note that for Ninja, the gap narrows at higher resolutions, suggesting that resolution scaling can partially overcome this architectural bias.
>
> We hope our responses address the Reviewer's requested changes adequately; we are happy to answer any further questions.
>
> **References**
>
> Raphael Trumpp, Ansgar Schäfftlein, Mirco Theile, and Marco Caccamo. Impoola: The power of average pooling for image-based deep reinforcement learning. Reinforcement Learning Journal, 6:1025–1047, 2025.

---

### Review · Reviewer_SLHY · 2026-03-27

**Summary Of Contributions:**

The paper studies the effect of input resolution in pixel-based deep RL and argues that the common practice of heavy downsampling is suboptimal. It shows that higher-resolution observations can improve performance and generalization, but only when the architecture scales properly. In particular, the authors identify a limitation of the standard Impala encoder, whose flattening operation leads to poor scaling with resolution, and propose a simple replacement using global average pooling (Impoola). Experiments suggest that this change allows the agent to benefit more consistently from higher-resolution inputs, especially in visually demanding environments.

**Audience:**

Yes

**Audience Explanation:**

The paper is well motivated and presents a valuable study of input resolution in pixel-based deep RL, offering a useful perspective for the community.

**Claims And Evidence:**

Yes

**Claims Explanation:**

1. The empirical message is clear: higher-resolution inputs can improve performance and generalisation when the architecture can effectively exploit the added visual detail.
2. The experimental study is comprehensive, covering multiple resolutions, model scales, and RL benchmarks, which makes the conclusions more convincing.
3. The paper highlights that a simple architectural change is highly effective in practice, especially in visually demanding environments.
4. The analysis goes beyond benchmark scores, with additional evidence such as gradient saliency that helps support the paper’s claims.

**Requested Changes:**

Although the paper has several strengths, I have a few major concerns.
1. Limited methodological novelty.
The paper’s main empirical message is interesting and useful, but the methodological novelty appears limited. The work mainly builds on existing architectural ideas rather than introducing a substantially new approach. In particular, the use of global average pooling is simple and effective, but it has already been explored in prior vision architectures and related RL studies [1, 2].

2. Overlap with prior work.
The paper is closely related to recent work such as Mind the GAP! [2], which also studies the role of global average pooling in addressing scale-related issues in pixel-based deep RL. This overlap limits the novelty unless the authors more clearly explain the main added value of the present paper beyond broader experimentation.

3. Limited baseline and experimental scope.
While the paper provides convincing evidence within its chosen setup, the study is still restricted to a narrow set of architectures (more are presented in [2]). and relatively limited resolution settings. As a result, it is unclear how broadly the conclusions extend beyond this regime. Comparisons with stronger or more diverse encoder baselines would make the claims more convincing.

4. Limited practical discussion.
While the paper shows that higher-resolution inputs can improve performance, it does not provide a sufficiently clear analysis of the associated computational trade-offs, such as training time, memory usage, and performance gain relative to added cost. A more explicit cost-performance comparison would make the practical implications of the paper much clearer.

---

> ### Author Response · Authors · 2026-04-10
> **Revised Manuscript**
>
> We thank Reviewer SLHY for their thorough reading of our manuscript and their valuable feedback. We incorporated the requested changes in the revised manuscript as follows:
>
> - **Points 1 & 2 - Methodological novelty and prior work:**
> The methodological contribution of this work lies in the introduction of scaling image resolution as a novel concept for deep RL, whereas prior scaling work has focused almost exclusively on network size. We introduce image resolution as a critical, yet largely neglected, hyperparameter in the training pipeline. Mind the GAP (Sokar & Castro, 2025) can be seen as a parallel work to Impoola (Trumpp et al., 2025); both identify GAP as beneficial for parameter scaling, but study it at a fixed conventional resolution. Our work goes a step further by introducing image resolution itself as a novel, independent variable.  Our results show that the Impoola architecture, in particular, can benefit significantly from higher resolutions (e.g., yielding a 28% performance improvement over Impala at their respective best conditions). We have revised the paper to make this distinction clearer.
>
> - **Point 3 - Baseline and experimental scope:** In the revised paper, we have added SoftMoE (Sokar et al., 2025) as a third architecture (see new Appendix B.1, Figure 14). This is a tokenization-based approach that, like Impoola, avoids flattening the spatial feature maps. The results support our thesis: SoftMoE also benefits from higher resolution more than Impala, confirming that architectures that handle spatial features in a structured way can leverage additional visual detail, while Impala's flattening creates a bottleneck. Impoola with GAP still performs best among the three, suggesting it remains the most effective approach in this setting.
>
> - **Point 4 - Practical discussion of computational trade-offs:** We are thankful for this valuable suggestion. In the revised manuscript, we have added Appendix B.3 (Figure 21), which shows the computational cost of resolution scaling, including total training time and VRAM usage across resolutions and width scales. For reference, increasing from 64x64 to 112x112 results in approximately 2.7x longer training time. Impoola and Impala have essentially identical training times, since the cost of the linear layer is negligible compared to that of the convolutional layers. We have also added a discussion of these costs in the Limitations section (Section 6).
>
> We hope our responses address the reviewer's requested changes adequately; we are happy to answer any further questions.
>
> **References**
>
> Ghada Sokar and Pablo Samuel Castro. Mind the GAP! the challenges of scale in pixel-based deep reinforcement learning. In The Thirty-ninth Annual Conference on Neural Information Processing Systems, 2025.
>
> Ghada Sokar, Johan Samir Obando Ceron, Aaron Courville, Hugo Larochelle, and Pablo Samuel Castro. Don’t flatten, tokenize! unlocking the key to softmoe’s efficacy in deep RL. In International conference on learning representations, 2025.
>
> Raphael Trumpp, Ansgar Schäfftlein, Mirco Theile, and Marco Caccamo. Impoola: The power of average pooling for image-based deep reinforcement learning. Reinforcement Learning Journal, 6:1025–1047, 2025.

---

> > ### Author Response · Authors · 2026-04-27
> >
> > **Small update for Point 3:** We used the time since our last update to extend our initial SoftMoE results to include all resolutions and updated Figure 14 accordingly.

---

> > > ### Comment · Reviewer_SLHY · 2026-07-15
> > >
> > > Thank you for the additional architecture and efficiency analyses.
> > >
> > > I have two remaining questions:
> > >
> > > First, while I acknowledge the contribution of systematically treating resolution as an independent scaling axis and relating its benefits to encoder design, describing resolution scaling as a “novel concept” in deep RL seems too strong / an overclaim. Prior work has already discussed visual resolution as an important hyperparameter and explored architectures for higher-resolution RL inputs (see two papers below). I suggest positioning the novelty more narrowly around the systematic scaling analysis and the interaction between resolution and encoder architecture. The abstract’s framing of resolution as a “critical yet overlooked variable” appears more accurate. This is primarily to do with framing rather than challenging the contributions.
> > >
> > > Second, what is the practical upper limit of resolution scaling? The current experiments cover a relatively narrow range. Does performance eventually saturate or degrade, and is this limit mainly determined by the architecture, the environment’s native rendering resolution, or computational cost? A wider resolution study, or at least a discussion of these limiting factors, would better characterize the scope of the proposed scaling axis.
> > >
> > >
> > > References:
> > >
> > > Leibo et al. (2018), Psychlab: A Psychology Laboratory for Deep Reinforcement Learning Agents.
> > >
> > > Choromanski et al. (2021), Unlocking Pixels for Reinforcement Learning via Implicit Attention.

---

> > > > ### Author Response · Authors · 2026-07-17
> > > >
> > > > Dear Reviewer SLHY,
> > > >
> > > > Thank you for reading our latest revision! Based on your questions, we are currently working on a revision of our manuscript, but wanted to give a short answer to your questions for the time being:
> > > >
> > > > - **Q1:** We agree that "novel concept" is not a fitting wording; note that this wording appeared only in our rebuttal comment here, while the manuscript itself uses the framing of a "critical yet overlooked variable". We will check again that the revised manuscript consistently uses this framing and positions its contribution around the systematic scaling analysis and the interaction between resolution and encoder architecture. In addition, thanks for pointing out the two references. We will also add Leibo et al. (2018) and Choromanski et al. (2021) to the Related Work section, as both are indeed relevant prior discussions of resolution in RL.
> > > >
> > > > - **Q2**: The main reason we did not go beyond R=112 for the time being was VRAM: as shown in Figure 23, training at R=112 with $\tau=3$ already requires close to 30 GB, near the limit of our A100 40GB GPUs (our implementation loads the full batch directly to VRAM for a big speedup). To better understand the upper limit, we have found a way to start a scaling experiment at even higher resolutions, but due to the high compute requirements for a single environment. Independent of these results, we note that the computational cost of resolution scaling has been discussed in Section 6 and Appendix B.3, and we consider it a practical rather than a fundamental limit. We will update the discussion to better reflect such practical limits.
> > > >
> > > > We hope to provide the updated manuscript in the next few days since we are waiting for the new scaling experiment to finish.
> > > >
> > > > With best wishes,
> > > >
> > > > The Authors

---

> > > > > ### Comment · Reviewer_SLHY · 2026-07-21
> > > > >
> > > > > Thank you. Looking forward to seeing the revision.

---

### Review · Reviewer_J5tB · 2026-06-15

**Summary Of Contributions:**

## Contributions:
1. In Procgen environments with changing rendering resolutions, by replacing the flatten layer after the vision backbone with a global average pooling (GAP) layer, deep RL can obtain a higher performance gain from high resolution.
2. They proved the above conclusion with rich experiments by changing various dimensions, such as the agent environment, rendering resolution, and network width.
3. Furthermore, they used analysis methods like gradient saliency and dormant neurons to analyze why the GAP layer benefits more from high resolution than the Flatten layer.


## Strengths:
1. The logic of this work makes sense: original deep RL did not consider changes in resolution. When the resolution changes, the parameters of the entire network will explode due to the Flatten layer. To solve this problem, they introduced the GAP layer from resolution-independent frameworks.
2. Rich experiments and representation analysis: They not only conducted rich experiments to fully prove their conclusion, but also did not stop at the surface. They further analyzed the reasons why the GAP layer wins through representation analysis.


## Weaknesses:
1. The argument that GAP is a fundamental and general solution to visual scaling might be too strong: Is extracting performance gains from higher-resolution environments merely an architectural parameter problem, and how well does GAP generalize to other architectures?
2. The analysis could go one step deeper: at higher resolutions, why does a larger depth help Impala in Section 4.1? Also, in the analysis of Section 5, the model's layers are not differentiated. Where exactly did the GAP layer help the most?

(I will elaborate on Weaknesses in my explanations below)

**Audience:**

Yes

**Audience Explanation:**

The paper challenges a prevailing assumption in the RL community that low-resolution inputs are sufficient, an assumption that has historically guided the design of standard network architectures. The authors systematically demonstrate that high-resolution inputs can actually lead to better performance, particularly in environments that require the precise observation of small entities.


Furthermore, the authors clearly identify that traditional architectures like Impala fail to effectively digest this rich visual information at higher resolutions. Through a comprehensive series of experiments and representation analyses, they convincingly show that introducing a GAP layer enables the model to achieve significant performance gains under high-resolution inputs. Given the clear motivation, practical problem identification, straightforward method, and solid empirical analysis, researchers working on visual deep RL will certainly find this work interesting and valuable.

**Broader Impact Concerns:**

None.

**Claims And Evidence:**

No

**Claims Explanation:**

While the empirical evidence presented is solid, I selected "No" because I have two concerns regarding the current manuscript: First, the core claims regarding GAP's role in visual scaling are significantly overstated. Second, I expect a deeper and more detailed mechanistic analysis to fully explain the internal learning dynamics.


Fundamentally, when rendering resolution increases while keeping the architecture fixed, the relative receptive field of the convolutional filters decreases. Consequently, the features extracted by the visual backbone inherently represent lower-level details. Building on this premise, my two main points are elaborated below:


## The fundamental and general nature of GAP:


- **Fundamenta**l:
  - The paper itself notes (Section 4.1) that simply increasing network depth can alleviate the issue. Following this logic, it is worth asking whether the problem could be solved simply by ensuring that the relative receptive field of the backbone's output features remains fixed as the resolution scales (e.g., by increasing network depth or enlarging kernel sizes).
  - Furthermore, in architectures like DrQ-v2 (designed to tackle high-resolution inputs in continuous domains like MuJoCo), a feature bottleneck is artificially introduced (compressing CNN outputs to a 50-dimensional vector before the MLP). Functionally, GAP acts as a similar structural bottleneck by aggressively compressing spatial dimensions into channels. If we artificially add a feature bottleneck into the standard Impala architecture (e.g., compressing the Flattened output to a much smaller dimension before the main MLP), would that also achieve similar performance gains?


- **Generalization**: The authors compare GAP with SoftMoE to prove its superiority. However, to strongly claim generalizability, it remains unclear whether replacing Flatten with GAP would still yield benefits in other modern architectures that already employ a structural feature bottleneck (like DrQ-v2) after the visual backbone.


## Deeper analysis of depth vs. GAP


The mechanistic explanation of why GAP benefits from high resolution lacks sufficient granularity.
- The success of using a larger depth is intuitive: it naturally restores the relative receptive field to a level comparable to low-resolution inputs while simultaneously capturing finer details from the high-resolution input.
- In contrast, GAP does not alter the receptive field at all. Therefore, equating both larger depth and GAP simply as "methods that reduce parameters compared to Flatten" oversimplifies their entirely distinct internal mechanics.


Furthermore, while Section 5 provides representation analysis (e.g., dormant neurons), it does not differentiate between network layers. After introducing GAP, does it force the CNN layers to extract more complex features at higher resolutions? Or does the CNN remain the same, and GAP simply provides a cleaner signal that allows the subsequent MLP to learn a better policy? A layer-wise analysis is necessary to clarify this.

**Requested Changes:**

- **Tone down the claims and explore other architecture parameters (Critical)**:
Please discuss or evaluate whether scaling well to higher resolutions can also be achieved by adjusting other architecture parameters (e.g., increasing convolution depth, enlarging kernel sizes, or explicitly adding a feature bottleneck after the vision backbone) rather than just replacing the Flatten layer. Acknowledging that other architectural adjustments might achieve similar results does not diminish the value of this work.


- **Generalization to other architectures**:
Please discuss whether replacing the Flatten layer with GAP would still work in other network architectures, particularly those that already incorporate a feature bottleneck by design, like what DrQ-v2 does.


- **Deeper mechanistic analysis**:
Provide a slightly more detailed analysis of how Conv depth and GAP actually work under the hood. For example, explicitly distinguish their different impacts on the receptive field, and consider providing a layer-wise breakdown for the dormant neuron analysis.

---

> ### Author Response · Authors · 2026-06-24
>
> Dear Reviewer J5tB,
>
> Many thanks for your valuable suggestions. We are particularly thankful for your remark about testing other modifications for Impala as well, since we think we have found some interesting aspects in this regard. We have revised our manuscript, using a *green font* this time to highlight our following changes:
>
> - **Point 1 - Claim for Impoola/GAP**: We have taken your feedback into account and adjusted our wording because we never intended to overemphasize GAP's role in visual scaling but to challenge the prevailing standard in general. As such, we have revised our claims in a more general and appropriate way that also reflects our new findings (see points below).
>
> - **General - Improved description for Impala architecture**: Our architecture description was updated to emphasize that Impala includes a Linear projection layer $z=We+b$ that serves as an output bottleneck ($z$ is the output vector, $e$ is the intermediate representation obtained by GAP/fattening of the feature maps $m$). We clarify why increasing R increases the parameter count in this layer quadratically in Impala, even when the output is set to a moderate value of $d(z)=256$. We have followed the reviewer’s suggestions and explained the effect of the receptive field with more care. We calculated the theoretical absolute receptive field (see Appendix B.4) and found that r=141 for the Impala base does, in theory, cover the whole input image, even for R=112 (note that the Impala network is, at least by deep RL standards, a quite deep network with 15 Conv2d layers).
>
> - **Point 2 - Generalization to other architectures with bottleneck layer:** We have clarified in the manuscript that there is already a linear projection layer present in Impala and Impoola (see above), which serves as a bottleneck layer, although it has a higher dimensionality, $d(z)=256$, compared to $d(z)=50$ in DrQ-v2.
>
> - **Points 1 & 2 - Impala variants**: Based on your feedback, we have included results for several modifications for Impala that are designed to compensate for the receptive field coverage when increasing R, or addressing the parameter explosion in the Linear projection layer:
>   - *Impala w/ Downsampling*: Increasing the kernel and stride in the first layer/max-pooling to restore a similar image coverage rate of the receptive field at higher resolutions.
>   - *Impala w/ d(z) = 50 (Bottleneck)*: As suggested, we set the output dimension of the Linear projection layer to a low value of d(z) = 50 (same value as DrQ-v2) to introduce a stronger information bottleneck.
>   - *Impala w/ DrQv2-Style (Bottleneck)*: Same as above, but with LayerNorm and Tanh activation to mimic the exact bottleneck from DrQ-v2.
>   - *Impala w/ 4 x ConvSeq*: As previously included, we describe the effect on the absolute receptive field in more detail.
>   - *Impala w/ Kernel (5,5)*: Increasing the Conv2d kernel size in all layers to (5,5), which is a rather unusual choice but increases the theoretical absolute receptive field.
>   - *Summary*: While the deeper network with 4 x ConvSeq performs well (as shown in the previous version), the bottleneck variants also yield interesting and improved results with visual scaling. Our results show that Impoola with GAP still outperforms these new variants which makes us think that GAP is the most compelling way to profit from visual scaling since it combines the best aggregated performance with practical aspects such as a simple network implementation and parameter count independent from the image resolution. However, we believe that this further evidence that other modifications (in addition to our SoftMoE results) also allow to unlock performance gains at higher resolutions strengthens our overall claim/intention to motivate others to treat input resolution as an important, tunable variable.
>
> - **Point 3 - Mechanistic analysis**: Our results include a per-layer discussion of dormant neurons now. See the full details in Appendix B.1.3. Thank you for this valuable idea because we found the interesting fact that the dormant neurons are particularly accumulated in the deepest Conv2d layers, i.e., in the first ConvSeq_0 sequence, when the layer is not connected to the higher layers by the residual connection. We suspect that this finding indicates gradient issues in these layers, which may arise from the high parameter count in the Linear projection layer.
>
> We hope our responses address the Reviewer's requested changes adequately and we are happy to answer any further questions.

---

> ### Author Response · Authors · 2026-06-29
>
> Dear Reviewer J5tB,
>
> Just a quick note that we extend the following aspect in our newest revision:
> - **Points 1 & 2 - Impala variants:** In addition to the results with Impala, we also tested Impoola with a further reduced bottleneck layer (from d(z)=256 to d(z)=50) now. We find that this has a slight regularizing function for Impoola, but the results are very close to the original Impoola results.

---

### Author Response · Authors · 2026-07-01
**Overview of Changes**

Dear Reviewers, Dear Action Editor,

Since we have updated our original manuscript several times during the reviewing process, we would like to provide a brief overview of the key changes and related findings in our latest revision:
- Extended experimental scope: Many new results including SoftMoE + several Impala variations (+ Impoola w/ stricter bottleneck); we find that SoftMoE and Impala with a stricter bottleneck layer also profit from visual scaling, but Impoola performs best plus has several practical advantages (simple implementation, low parameter count, etc.).
- Practical discussion of computational trade-offs, e.g., training times and VRAM usage.
- Per-layer analysis of dormant neurons, where we find that dormant neurons particularly accumulate in the deepest layers (ConvSeq0) in Impala when scaling the resolution.
- Improved description of the Impala network; formal definition of the Linear projection layer and its dependency on resolution.
- Others: Improved wording for the claim to be more in line with the goal of the work to establish input resolution as a tunable parameter; more precise analysis of anomalies in the results.

As such, we would also like to thank everyone for their thorough feedback, which has helped us improve our work!

With best wishes,

The Authors